# Foldable SuperNets: Scalable Merging of Transformers with Different Initializations and Tasks

**Edan Kinderman**                                        *kinderman7@gmail.com*
*Electrical and Computer Engineering Department, Technion*

**Itay Hubara**                                           *itayhubara@gmail.com*
*Habana Labs – An Intel company*

**Haggai Maron**                                          *haggaimaron@gmail.com*
*Electrical and Computer Engineering Department, Technion*
*NVIDIA Research*

**Daniel Soudry**                                         *daniel.soudry@gmail.com*
*Electrical and Computer Engineering Department, Technion*

**Reviewed on OpenReview:** *https://openreview.net/forum?id=6FqwLestHv*

## Abstract

Recent methods aim to merge neural networks (NNs) with identical architectures trained on different tasks into a single multi-task model. While most works focus on the simpler setup of merging NNs initialized from a common pre-trained network, we target the harder problem of merging large transformers trained on different tasks from distinct initializations. We show that traditional merging methods fail catastrophically in this setup, while Knowledge Distillation (KD) achieves much better results, though at a higher cost. However, KD is data-inefficient, as it does not exploit the original models' weights. To solve this, we introduce "Foldable SuperNet Merge" (FS-Merge), which trains a SuperNet containing the original models (with frozen weights) using a feature reconstruction objective. After training, the SuperNet is folded back to the size of a single original model. FS-Merge is simple, data-efficient, has a computational cost comparable to KD, and is proven to have superior expressiveness compared to traditional merging methods on MLP models. It achieves SOTA results when tested on MLPs and transformers across various sizes, tasks, modalities, and distribution shifts, especially in low-data scenarios[1].

## 1 Introduction

Practitioners frequently train identical neural architectures for various tasks and share these models online, while the original training data is often unavailable due to privacy, proprietary, or other concerns. This led to an increased interest in the field of model merging (Akhlaghi & Sukhov, 2018; Goddard et al., 2024), which aims to combine the weights, and sometimes the features, of several models into a single new model (Figure 1). This approach could allow the merging of multiple single-task models into a single multi-task model (Matena & Raffel, 2022; Ilharco et al., 2023), eliminating the need to store and run multiple models via Mixture-of-Experts Shazeer et al. (2017) or ensembles (Ganaie et al., 2022).

Most existing merging methods have a strong restriction: they assume that the models were initialized from the same pre-trained model and subsequently fine-tuned. This encourages the models to stay aligned (Ainsworth et al., 2023) and also to remain closer in the weight space Ilharco et al. (2023), and therefore easier to fuse, for example by simply averaging their weights (Wortsman et al., 2022). However, this restricts the

---

[1]Code is available at `https://github.com/idankinderman/fs_merge`.

Table 1: Merging a pair of ViT-B-16, fine-tuned on Cars and CIFAR10, using 100 original images and 800 augmented images from each dataset. The test accuracy is averaged on both tasks.

| Method | Accuracy | Method type |
|---|---|---|
| Ensemble | 89.27 | - |
| Random guess | 5.25 | - |
| Average | 5.56 | Averaging-based |
| SLERP | 4.80 | Averaging-based |
| RegMean | 6.58 | Averaging-based |
| Opt | 6.32 | Alignment-based |
| Distillation | 75.81 | Training-based |
| FS-Merge (Ours) | 85.83 | Training-based |

ability to merge models that do not share the same initialization. For example, consider the task of merging the weights of two unrelated models from an online repository (e.g., Hugging Face or GitHub). These models were likely not fine-tuned from the same initial model, making most existing merging techniques inapplicable.

To overcome this, several studies have explored merging differently initialized models using alignment-based methods (Entezari et al., 2022; Verma & Elbayad, 2024; Ainsworth et al., 2023; Stoica et al., 2024). This family of methods is more resource-intensive compared to averaging-based methods. However, these approaches still use relatively simple merging rules and thus struggle with more complicated tasks such as merging transformers (Stoica et al., 2024). For a detailed discussion of related work, refer to Appendix A.1.

To illustrate the limitations of these methods, we pre-trained two Vision Transformers (ViTs) (Dosovitskiy et al., 2021) with *different initializations* on ImageNet-1k (Deng et al., 2009), fine-tuned each on separate tasks (Cars (Krause et al., 2013) and CIFAR10 (Krizhevsky et al., 2009)), and then merged them (Table 1). Traditional approaches like weight averaging, SLERP (Shoemake, 1985), and RegMean (Jin et al., 2023), as well as the alignment merging method designed for transformers, 'Opt' (Imfeld et al., 2023), all failed in this setting, resulting in a merged model with performance comparable to random guessing. Moreover, these methods show a significant accuracy gap compared to the model ensembling (Ganaie et al., 2022), a method that averages the model outputs. Note that the ensemble is not a valid merging method, as it uses the original models directly. These traditional merging methods rely on simple local merging rules, ensuring computational efficiency. However, they proved inadequate for our complex setting due to their simplicity. This pattern persisted across all other settings, tasks, and modalities we tested with transformers (see Section 3.2).

These results indicate that merging large transformers from diverse initializations demands stronger and more resource-intensive techniques, such as Knowledge Distillation (KD) (Ba & Caruana, 2014; Hinton et al., 2015). In the multi-task setting, KD refers to a single model trained to replicate the outputs of multiple models (Tan et al., 2019; Vongkulbhisal et al., 2019). And indeed, KD significantly outperforms the previous methods (Table 1), but still exhibits a large performance gap compared to the ensemble. Although KD shows promise, it often requires access to large parts of the original training dataset and labels (Clark et al., 2019; Liu et al., 2020) to achieve high accuracy, which may be problematic due to privacy or proprietary concerns. Moreover, it does not explicitly utilize the original models' weights, potentially overlooking valuable information.

**Our approach.** To leverage the hidden knowledge in the original weights and further improve accuracy beyond KD, we propose the "Foldable SuperNet Merge (FS-Merge)" method. This approach utilizing a "SuperNet"— a network in which the original models are embedded with frozen parameters. The SuperNet weights are then trained to minimize either local or global feature reconstruction loss. After training, the SuperNet weights are folded to produce a model with the same size as the original models (Figure 2). Notably, this means that the additional computational cost occurs only once, during merging.

FS-Merge is simple, data-efficient, and we prove that it has superior expressivity compared to the baselines (Appendix H). Like other methods (Jin et al., 2023; Ainsworth et al., 2023), it requires only an unlabeled

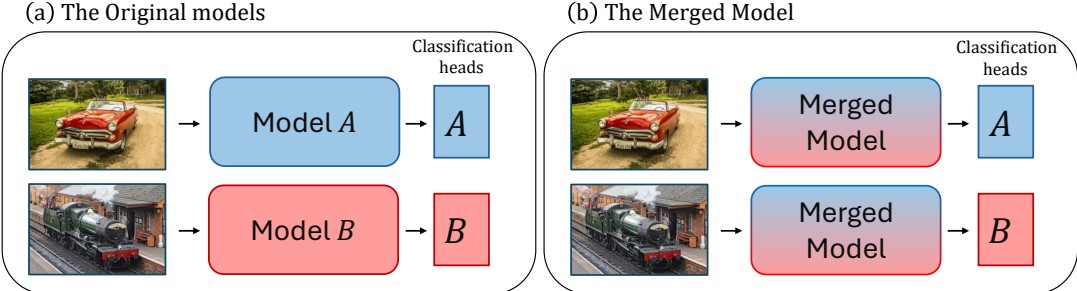

Figure 1: **The model merging setting**. (a) Consider models $A$ and $B$ trained from different initializations and on different tasks, which create features processed by a classification head for predictions. (b) Merging methods fuse the models into a new model of the same size while leaving the classification head untouched. The merged model generates features applicable to all tasks.

fraction of the original training data. We demonstrate its effectiveness by achieving SOTA results across diverse scenarios, architectures, model sizes, tasks, modalities, and distribution shifts.

We prove that FS-Merge offers greater expressiveness compared to traditional merging methods. Moreover, by leveraging the original weights, FS-Merge outperforms KD, especially in data-limited regimes, significantly reducing the gap with the ensemble (Table 1), and even surpassing it in some cases. While this work focuses on transformers, FS-Merge's versatility allows it to be extended to other architectures such as RNNs (Sherstinsky, 2020) and CNNs (He et al., 2016). This adaptability sets FS-Merge apart from alignment-based methods, which are often designed for specific architectures.

## 2 Method

**Problem formulation.**   For simplicity, we outline the merging problem for two models $A$ and $B$ with identical widths, trained on distinct tasks with unique initializations and separate classification heads (Figure 1a). Using unlabeled subsets of each task's training set, $D^A$ and $D^B$, our goal is to construct a new model with the same architecture, that minimizes the losses for tasks $A$ and $B$. Be aware that the classification heads are not merged, meaning we retain a separate head for each task (Figure 1b). Note that FS-Merge can be easily extended to merge any number of models or models with varying widths. We denote $D = D^A \cup D^B$.

In this section, we introduce FS-Merge, our proposed approach, first for merging MLPs and then for the more complex task of merging transformers.

### 2.1   Warmup: merging Multi-Layer Perceptrons

Let us consider the $l$-th layer in the Multi-Layer Perceptron (MLP) model $A$. The features at this layer, denoted by $f_l^A \in \mathbb{R}^{d_l}$, can be expressed as follows:

$$z_l^A = W_l^A f_{l-1}^A + b_l^A \, , \, f_l^A = \sigma(z_l^A) \, . \tag{1}$$

Here, $W_l^A \in \mathbb{R}^{d_l \times d_{l-1}}$ and $b_l^A \in \mathbb{R}^{d_l}$ are the weights and biases of the current linear layer, respectively. $f_0^A = x$ denotes the MLP input, $d_l$ denotes the width of the $l$-th layer, $\sigma$ represents a non-linear element-wise activation function, and $z_l^A \in \mathbb{R}^{d_l}$ are the pre-activation features.

Our method has two versions: local FS-Merge and global FS-Merge. Each has its own Foldable SuperNet and reconstruction optimization problem, neither requiring true labels. The parameters learned during FS-Merge optimization are highlighted in red, and the notation $\tilde{f}$ represents a feature's reconstruction attempt of the Foldable SuperNet.

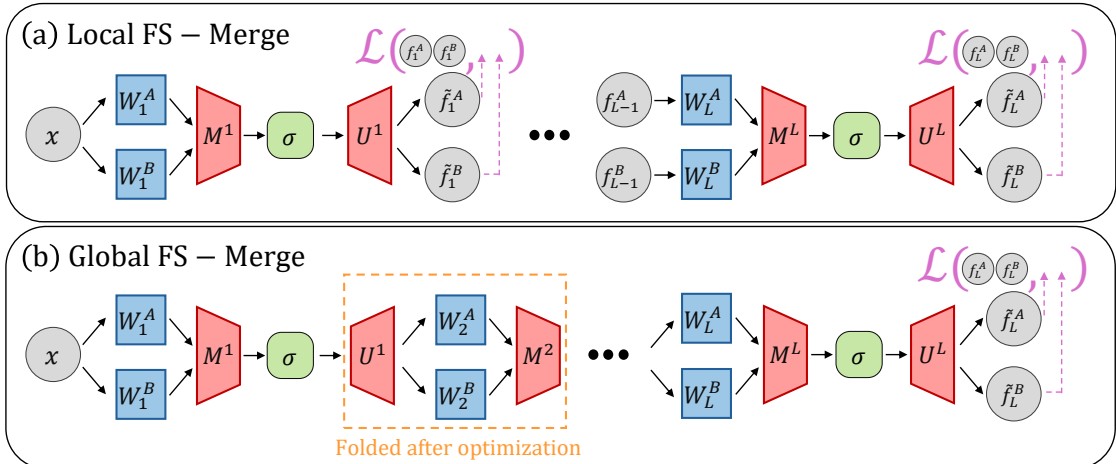

Figure 2: **The FS-Merge for MLP.** (a) Local FS-Merge: $M_l$ and $U_l$ are optimized separately for each layer $l$ to reconstruct $f_l^A || f_l^B$. (b) Global FS-Merge: composing the $M$ and $U$ matrices for each one of the $L$ layers of the MLP to reconstruct $f_L^A || f_L^B$, the features before the classification head. In both versions, after optimization, the Foldable SuperNet is folded to create the merged model. Red represents the SuperNet weights, blue represents the original frozen weights, gray represents features, and green represents the activation function.

**Local FS-Merge.** In the case of merging the $l$-th linear layers of models $A$ and $B$, the local Foldable SuperNet (Figure 2a) is defined as follows:

$$\tilde{f}_l(z_l^A, z_l^B) = U_l \sigma(M_l(z_l^A || z_l^B)), \tag{2}$$

The input is a concatenation of the original pre-activation features $z_l^A || z_l^B \in \mathbb{R}^{2 \cdot d_l}$. $M_l \in \mathbb{R}^{d_l \times 2 \cdot d_l}$ ("Merge") is used to merge the original features into a lower-dimensional space (from $2 \cdot d_l$ to $d_l$), and $U_l \in \mathbb{R}^{2 \cdot d_l \times d_l}$ ("Unmerge") is used to approximately reverse the merge operation, attempting to reconstruct the original post-activation features $f_l^A || f_l^B \in \mathbb{R}^{2 \cdot d_l}$ with $\tilde{f}_l \in \mathbb{R}^{2 \cdot d_l}$. Another way to express $\tilde{f}_l$ is by $\tilde{f}_l^A || \tilde{f}_l^B$.

For optimization, $D$ is used to extract the models features $f_l^A$, and $f_l^B$. Then, $M_l$ and $U_l$ are optimized separately for each layer $l$, on the following reconstruction optimization problem:

$$M_l^*, U_l^* = \underset{M_l, U_l}{\operatorname{argmin}} \, \mathbb{E}_{x \sim D} \, \left\| f_l^A || f_l^B - \tilde{f}_l(z_l^A, z_l^B) \right\|_2^2. \tag{3}$$

**Global FS-Merge.** In the global version (Figure 2b), the Foldable SuperNet is created by composing the $M$ and $U$ matrices for each one of the $L$ layers of the MLP. In the forward pass, the $l$-th layer of the Foldable SuperNet uses the reconstructed pre-activation features of the previous layer $\tilde{z}_l^A, \tilde{z}_l^B$ as inputs. Observe that this does not include the classification head, which is not being merged. Then, all those matrices are optimized together on the following global optimization problem:

$$M_1^*, U_1^*, ..., M_L^*, U_L^* = \underset{M_1, U_1, ...}{\operatorname{argmin}} \, \mathbb{E}_{x \sim D} \, \left\| f_L^A || f_L^B - \tilde{f}_L(x) \right\|_2^2, \tag{4}$$

where $f_L^A \in \mathbb{R}^d$ are features from the last representation layer ($L$-th layer) of model $A$, applied the input $x$. The output of the Foldable SuperNet is defined as $\tilde{f}_L \in \mathbb{R}^{2 \cdot d}$, which attempts to reconstruct $f_L^A || f_L^B \in \mathbb{R}^{2 \cdot d_L}$. As we will demonstrate later, the global problem allows us to deal with more complicated architectures such as transformers. Note that we have a slight abuse of notation, as $\tilde{f}_L$ denotes reconstructed features from both the local and global FS-Merge.

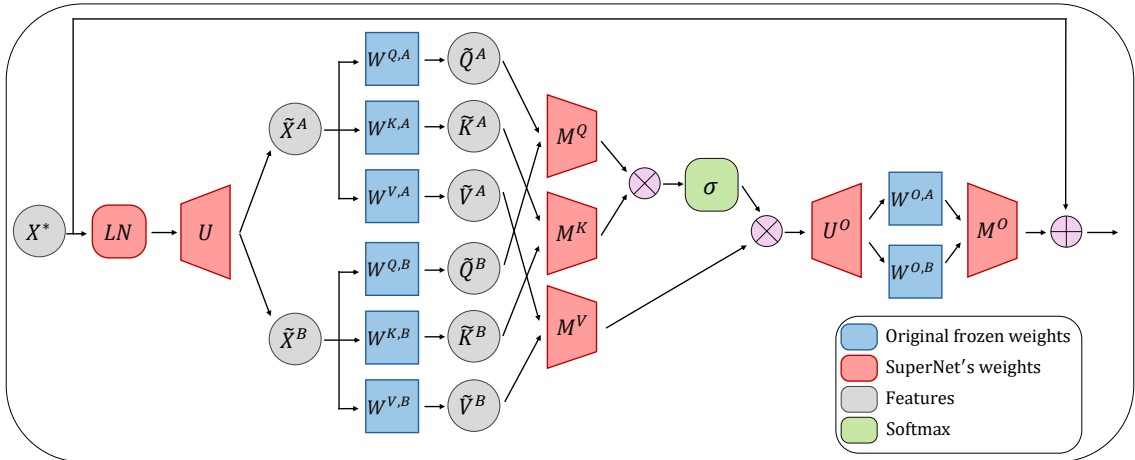

Figure 3: **The Foldable SuperNet for merging two attention blocks ($A$ and $B$).** Only the red components are trained. $M$ matrices are used to merge the models' features into merged features ($X^*$), and $U$ are applied to the merged features to reconstruct the original ones ($\tilde{X}^A, \tilde{X}^B$). After training, the Foldable SuperNet is folded to produce a merged attention block with the same size as the original blocks.

**Folding.** After optimizing the $M$ and $U$ matrices in all layers (using the local or the global version), we can "fold" the Foldable SuperNet in order to create the merged model. The "folding" operation is defined as follows:

$$W_l^* = M_l^* \begin{pmatrix} W_l^A & 0 \\ 0 & W_l^B \end{pmatrix} U_{l-1}^*, \; b_l^* = M_l^* \begin{pmatrix} b_l^A \\ b_l^B \end{pmatrix}. \tag{5}$$

Where $U_0 = \begin{pmatrix} I \\ I \end{pmatrix}$. Intuitively, this folding operation creates a merged model which "under the hood" reconstructs the original features from the previous layer (using $U_{l-1}$), applies the original weights, and then merges those features again ($M_l$). This occurs with the same complexity as each of the original models, since $W_l^* \in \mathbb{R}^{d_l \times d_{l-1}}$.

**Initialization.** The Foldable SuperNet can be initialized in various ways (full details in Appendix G.1). In the simplest approach, the $M$ and $U$ matrices may be initialized randomly. We found that the best approach is to initialize $M$ and $U$ using the "first" initialization method (as described in Eq. 6), where only the weights of the first model are selected. This implies that, if the weights are folded at this point, we obtain $W_l^* = W_l^A$. For an ablation study in the matter, see Appendix G.1 and Appendix G.2.

$$M_l = \begin{pmatrix} I & 0 \end{pmatrix}, \; U_l = \begin{pmatrix} I \\ I \end{pmatrix} \tag{6}$$

**Relation with ZipIt (Stoica et al., 2024).** This "folding" operation was proposed by ZipIt (Stoica et al., 2024), which is closely related to our work, and inspired it. This method also merges models from various initializations and tasks, targeting MLPs and CNNs. In the MLP context, ZipIt represents a special case of FS-Merge, where $M$ is chosen to average highly correlated pairs in a hard-coded way, and $U = 2M^\top$. For other layers, such as skip connections and normalizations, ZipIt employs various heuristics that cannot be easily extended to transformers and may harm performance. For more details, see Appendix A.2.

## 2.2 Merging transformers

Merging transformers (Vaswani et al., 2017) is a much more challenging problem than merging MLPs, due to their larger scale and more complicated structure. Creating a naively Foldable SuperNet as described in the MLP Section 2.1 is not feasible for transformers because it fails to account for their skip connections, layer normalization (Ba et al., 2016), and multi-head attention.

To tackle these issues, we develop a new Foldable SuperNet architecture, and train it using the global objective (Eq. 4). We found that the global objective is essential for merging transformers, as training the SuperNet locally (Eq. 3) significantly reduces the accuracy of the resulting merged model (reasons for this are discussed in Appendix G.3).

**Foldable SuperNet for attention blocks.** In this section, we briefly describe the Foldable SuperNet for the attention block (Figure 3), as it is the most complex component of the transformer. For full technical details on the SuperNet for this block, as well as for the other transformer blocks (MLP and pre-processing), please refer to Appendix B.1. The optimized parameters at this stage are highlighted in red, with $\tilde{X}$ represents a reconstruction attempt of our method, and $X^*$ represents the merged features.

Suppose there are two transformers trained on distinct tasks $A$ and $B$. The Foldable SuperNet for their attention blocks receives as input $X_l^* \in \mathbb{R}^{T \times d}$, the merged features from the previous SuperNet, where $T$ is the sequence length and $d$ is the embedding size. It then applies a layer norm (which is also learned, as proposed in other merging works (Jordan et al., 2023)), and uses a learned matrix $U_l \in \mathbb{R}^{d \times 2 \cdot d}$ to reconstruct the original model features $\tilde{X}_l^A || \tilde{X}_l^B \in \mathbb{R}^{T \times 2 \cdot d}$.

Next, the Foldable SuperNet calculates the queries, keys, and values for $A$ and $B$ using the original frozen weights. The queries of both models are then concatenated and merged using a learnable matrix $M_l^Q \in \mathbb{R}^{2 \cdot d \times d}$, and similarly for the keys and values with $M_l^K, M_l^V$. Following this, the Foldable SuperNet performs multi-head attention with the merged queries, keys, and values; uses $U_l^O \in \mathbb{R}^{d \times 2 \cdot d}$ to reconstruct the original multi-head attention output; applies the aggregation frozen weights of the original models; uses $M_l^O \in \mathbb{R}^{2 \cdot d \times d}$ to compress it once more; and applies the skip connection. This process results in merged features of dimension $T \times d$, serving as the input for the subsequent Foldable SuperNet of the MLP block at layer $l$.

**Parameterizing $M$ and $U$.** Utilizing full-rank $M_l \in \mathbb{R}^{d_l \times n \cdot d_l}$ and $U_l \in \mathbb{R}^{n \cdot d_l \times d_l}$ for merging $n$ models introduces a number of parameters that increase quadratically with the layer width $d_l$. In large models like transformers, this leads to a very high demand for resources, hinders the optimization process, and cause overfitting. To mitigate this, we adopt a parameterization strategy akin to LoRA (Hu et al., 2022), using a sum of a low-rank matrix and a concatenation of diagonal matrices. For instance, in $M_l$:

$$M_l = M_l^{\mathrm{diag}} + M_l^1 M_l^2 \,. \tag{7}$$

When $r$ is the inner rank, $M_l^1 \in \mathbb{R}^{d_l \times r}$, $M_l^2 \in \mathbb{R}^{r \times n \cdot d_l}$, and $M_l^{\mathrm{diag}} \in \mathbb{R}^{d_l \times n \cdot d_l}$ is a concatenation of a $n$ diagonal matrices, each with $d_l$ learnable parameters. A similar structure is proposed for $U_l$. This ensures the number of learnable parameters is linear with the layer width $d_l$. We found that adding $M_l^{\mathrm{diag}}$ is crucial for FS-Merge, as it enables initializing the Foldable SuperNet with strong initializations such as "first" (Eq. 6).

**FS-Merge seq.** To address the high costs of merging a large number of models, we introduce a more efficient variant called FS-Merge Seq., which merges the models sequentially. It starts with the first two models, then continues by merging the resulting merged model with the third model, and so on. Full details can be found in Appendix B.4.

**Data and Augmentation.** This work addresses a realistic setting, involving a limited subset of unlabeled samples used for merging. For transformer merges, augmentations (Zhang et al., 2018) are employed to expand this subset, as commonly done in regular training. Appendix G.1 studies the effect of using augmentation on accuracy.

**Expressive power.** As shown in Section 3.2, FS-Merge significantly surpasses traditional merging methods. We explain this with theoretical results demonstrating its superior expressivity over the baselines.

**Theorem 2.1** (Informal)**.** *FS-Merge can exactly reproduce any target MLP model when merging models with sufficient rank conditions. Moreover, FS-Merge strictly generalizes previous merging methods like RegMean and Git Re-Basin, in the context of merging MLPs.*

See Appendix H for the formal statement and conditions.

Table 2: We merged pairs of MLPs, each initialized differently and trained on distinct halves of the **MNIST** dataset. These MLPs have a hidden width of 128 neurons, with the **number of hidden layers varying** from 1 to 6. Each experiment was replicated five times. We present the average **per-task accuracy** on the test set, along with the standard deviation.

| Model | Number of Hidden Layers | | | | |
|---|---|---|---|---|---|
| | 1 | 2 | 3 | 4 | 6 |
| Two models | $96.83 \pm 0.13$ | $96.51 \pm 0.19$ | $96.46 \pm 0.23$ | $95.86 \pm 0.40$ | $96.81 \pm 0.58$ |
| Ensemble | $94.70 \pm 0.95$ | $95.14 \pm 1.12$ | $95.73 \pm 0.22$ | $95.58 \pm 0.12$ | $95.78 \pm 0.73$ |
| average | $94.36 \pm 0.76$ | $85.90 \pm 4.46$ | $78.78 \pm 6.72$ | $61.76 \pm 6.36$ | $25.12 \pm 2.70$ |
| RegMean | $95.90 \pm 0.37$ | $92.97 \pm 2.71$ | $92.11 \pm 1.90$ | $87.79 \pm 3.77$ | $81.55 \pm 2.49$ |
| ZipIt | $96.35 \pm 0.17$ | $95.75 \pm 0.58$ | $95.43 \pm 0.50$ | $94.59 \pm 0.50$ | $94.03 \pm 2.27$ |
| Distillation | $93.35 \pm 1.07$ | $93.13 \pm 1.74$ | $93.71 \pm 0.39$ | $93.38 \pm 0.67$ | $90.79 \pm 1.89$ |
| FS-M | $95.89 \pm 0.03$ | $95.68 \pm 0.19$ | $95.37 \pm 0.33$ | $94.94 \pm 0.43$ | $94.89 \pm 0.89$ |
| FS-M, ZipIt init | $\mathbf{96.62 \pm 0.08}$ | $\mathbf{96.29 \pm 0.24}$ | $\mathbf{96.18 \pm 0.20}$ | $\mathbf{95.63 \pm 0.52}$ | $\mathbf{96.22 \pm 0.84}$ |

## 3 Results

We evaluate our method on MLPs (Section 3.1), Vision Transformers (Section 3.2), and Text Transformers (Section 3.3), and show it achieves SOTA results.

**Baselines.** We compared with "Original Models", representing the average accuracy of the models to be merged; and Ensemble (Ganaie et al., 2022), which averages the models outputs and then applies classification heads. Note that these are not valid merging methods as they use the original models directly. For legitimate merging techniques, comparisons were made with weight averaging (Wortsman et al., 2022); RegMean (Jin et al., 2023) which applies a closed-form linear regression solution to each layer; and distillation (Hinton et al., 2015) which trains a single model to mimic the pre-classification layer features. ZipIt (Stoica et al., 2024) averages highly correlated neurons, used only in the MLP experiments, as it is inapplicable for transformers. "Opt" (Imfeld et al., 2023), uses optimal transport for aligning transformers, and "SLERP" (Shoemake, 1985), utilized only on the ViT case, as they introduced for transformers.

**Data.** When merging the models, a small unlabeled subset of their original training data is used. In the case of merging ViTs, this subset is further extended using augmentations. For a fair comparison, all baselines use the same amount of original and augmented data.

**Metrics.** Following ZipIt (Stoica et al., 2024), we evaluate multi-task merged models using per-task and joint accuracy. Per-task accuracy is calculated by evaluating each task individually with its relevant classification head and reporting the mean accuracy across all tasks. Joint accuracy is calculated by making predictions for each task based on the maximum score across all classification heads and reporting the mean accuracy across tasks.

### 3.1 Warm-up: Merging Multi-Layer Perceptrons

**Setting.** MNIST (LeCun, 1998) was split into two subsets: images with labels 0–4 and images with labels 5–9. These subsets were further divided into training, validation, and test sets. Two MLPs were trained separately on these subsets, varying in the number of layers and widths, each initialized with a different seed. The goal was to merge these pairs of models, excluding the last linear layer, which serves as the classification head.

**FS-Merge.** Our method was tested in two variants: the local version of FS-Merge (Eq. 3), which trains a Foldable SuperNet for each layer independently; and FS-Merge ZipIt, which initializes the Foldable SuperNet's $M$ and $U$ as the solutions of ZipIt (Stoica et al., 2024), and then optimize them using the local version. **Data.** All merging methods, excluding "Average", use features from the original models. Thus, we sampled 64 images from each dataset's training set to generate the necessary features.

Table 3: Merging pairs of ViT-B-16 using 16 original images from each training set and 800 augmented images from each dataset. The per-task and joint accuracy on the test set are reported

| Merging Methods | DTD, EuroSAT | | CIFAR100, SVHN | | RESISC45, SVHN | | |
| --- | --- | --- | --- | --- | --- | --- | --- |
| | Per-task | Joint | Per-task | Joint | Per-task | Joint | Learnable Parameters |
| Original models | 81.55 | - | 91.19 | - | 95.12 | - | - |
| Ensemble | 78.64 | 74.11 | 88.26 | 57.44 | 93.21 | 75.07 | - |
| Average | 11.48 | 1.15 | 3.71 | 0.84 | 6.23 | 3.68 | ✗ |
| SLERP | 7.99 | 1.27 | 4.10 | 0.65 | 6.67 | 3.24 | ✗ |
| RegMean | 8.40 | 1.69 | 6.17 | 1.27 | 6.44 | 0.74 | ✗ |
| Opt | 4.51 | 0.75 | 4.52 | 0.88 | 7.01 | 2.07 | ✗ |
| Distillation | 57.31 | 52.61 | 62.93 | 48.12 | 66.18 | 63.16 | ✓ |
| Distillation, LoRA $r = 4$ | 57.23 | 51.83 | 58.81 | 49.83 | 62.17 | 57.04 | ✓ |
| Distillation, LoRA $r = 12$ | 55.07 | 48.28 | 59.06 | 47.59 | 64.16 | 58.66 | ✓ |
| FS-M, $r = 12$ | **62.64** | **57.42** | **66.50** | **53.98** | **73.77** | **69.26** | ✓ |

Table 2 presents the per-task accuracy on the test set, for merging MLPs trained on half of the MNIST dataset. We merged MLPs with 128 hidden widths, and hidden layers varying from 1 to 6. Each experimental condition was replicated five times with 5 different seeds. The same hyperparameters were used for all those experiments. Full information about the setting and hyperparameters are available in Appendix E.1.

Our results indicate that merging deeper models is more challenging, consistent with previous studies (Jordan et al., 2023). Employing the ZipIt initialization, our method establishes a new SOTA for both per-task and joint accuracy, outperforming ensemble in many cases, and nearly matches the accuracy of "Original Models". For results on more tasks and FS-Merge versions, see Appendix F.1

### 3.2 Merging Vision Transformers

Next, we evaluated our method on merging Vision Transformers (ViT) initialized differently and trained on distinct tasks, a much more challenging problem.

**Models and Data.** Several ViT-B-16 and ViT-L-14 models (Dosovitskiy et al., 2021; Touvron et al., 2021) were pre-trained on ImageNet-1K (Deng et al., 2009), using distinct random seeds. Then, each differently pre-trained ViT was fine-tuned on downstream tasks. Following (Ilharco et al., 2023), we fine-tuned on Cars (Krause et al., 2013), DTD (Cimpoi et al., 2014), EuroSAT (Helber et al., 2019), GTSRB (Stallkamp et al., 2011), MNIST (LeCun, 1998), RESISC45 (Cheng et al., 2017), SVHN (Netzer et al., 2011), CIFAR10, and CIFAR100 (Krizhevsky et al., 2009). For extended details regarding the setting refer to Appendix D.

In these experiments, the KD baseline aims to reconstruct the features from the last layer of the models to be merged, using the "first" initialization, meaning that the student model was initialized from the first model. We tested additional KD variants, such as a baseline that attempts to mimic the internal features of the models (Appendix G.5), and baselines where the student model is initialized using different merging methods (such as RegMean) before continuing training (Appendix G.2). Nevertheless, we found that our KD baseline, which uses only the last-layer features with the "first" initialization, outperforms all other variants. The KD LoRA is identical to the KD baseline, except that it trains only LoRA Hu et al. (2022) adapters with a low rank $r$ for each linear layer in the model.

**FS-Merge.** The global version of FS-Merge was used (Eq. 4), with the Foldable SuperNet's $M$ and $U$ parametrized as a concatenation of diagonal matrices plus low-rank matrices (Eq. 7). The Foldable SuperNet was initialized using the "first" initialization (Eq. 6), which showed the best performance (as demonstrated in Appendix G.1). FS-Merge Seq. (Appendix B.4) is a memory and compute-efficient variant of FS-Merge, designed for merging a large number of models. In Table 3, pairs of ViT-B-16 models fine-tuned on different

Table 4: Merging groups of 4 ViT-B-16 using 100 original and 1000 augmented images per training dataset (a total of 400 original images and 4,000 augmented images). We will denote: C = Cars, D = DTD, E = EuroSAT, G = GTSRB, M = MNIST, R = RESISC45, S = SVHN, C10 = CIFAR10, C100 = CIFAR100.

| Merging Methods | R, C10, S, G | | D, G, E, R | | C, M, C100, E | | Learnable Parameters |
|---|---|---|---|---|---|---|---|
| | Per-task | Joint | Per-task | Joint | Per-task | Joint | |
| Original models | 96.47 | - | 88.72 | - | 92.61 | - | - |
| Ensemble | 86.11 | 46.80 | 76.81 | 52.96 | 82.04 | 63.92 | - |
| Average | 5.40 | 1.04 | 3.66 | 1.35 | 4.55 | 0.38 | ✗ |
| SLERP | 5.55 | 1.04 | 4.88 | 0.87 | 7.27 | 0.86 | ✗ |
| RegMean | 6.38 | 0.61 | 4.78 | 0.61 | 5.60 | 0.52 | ✗ |
| Opt | 5.79 | 0.24 | 3.70 | 0.38 | 5.75 | 2.58 | ✗ |
| Distillation | 82.09 | 67.60 | 67.31 | 57.67 | 45.75 | 40.62 | ✓ |
| Distillation, LoRA $r = 4$ | 72.92 | 54.73 | 57.91 | 46.36 | 75.76 | 70.34 | ✓ |
| Distillation, LoRA $r = 12$ | 77.43 | 60.18 | 63.56 | 52.81 | 75.52 | 69.29 | ✓ |
| Distillation, LoRA $r = 24$ | 78.58 | 61.47 | 63.61 | 52.93 | 68.75 | 64.31 | ✓ |
| FS-M, $r = 24$ | **84.58** | 70.92 | 68.41 | 56.40 | **79.71** | **73.98** | ✓ |
| FS-M seq., $r = 16$ | 83.90 | **71.07** | **68.96** | **58.48** | 73.57 | 68.31 | ✓ |

Table 5: We merge pairs of ViT-B-16 models using 16 original and 800 augmented images per dataset. Accuracy is reported on the corrupted test sets of CIFAR100-C, with mean and standard deviation computed over the five severity levels.

| Corruption type | CIFAR100 ViT-B-16 | CIFAR100, RESISC45 | | CIFAR100, EuroSAT | |
|---|---|---|---|---|---|
| | | Distillation | FS-Merge | Distillation | FS-Merge |
| Snow | 71.9 ± 5.1 | 63.4 ± 5.5 | **71.2 ± 5.2** | 64.1 ± 5.2 | **69.6 ± 5.6** |
| Frost | 69.6 ± 8.8 | 61.2 ± 8.9 | **69.0 ± 8.9** | 62.2 ± 9.2 | **64.6 ± 10.9** |
| Fog | 75.7 ± 11.2 | 66.4 ± 13.1 | **75.2 ± 11.2** | 68.5 ± 12.4 | **74.4 ± 11.3** |
| Brightness | 82.8 ± 2.9 | 75.4 ± 3.6 | **82.4 ± 3.0** | 76.4 ± 3.4 | **81.8 ± 3.0** |
| Gaussian noise | 24.9 ± 16.8 | 22.8 ± 14.3 | **24.2 ± 17.0** | **22.4 ± 14.8** | 21.5 ± 15.8 |
| Shot noise | 33.7 ± 21.9 | 30.3 ± 18.6 | **33.0 ± 22.0** | **30.6 ± 19.2** | 30.0 ± 21.1 |
| Impulse noise | 30.0 ± 20.3 | 21.8 ± 15.9 | **28.6 ± 19.8** | 20.3 ± 17.2 | **26.3 ± 20.5** |
| Contrast | 69.3 ± 20.7 | 61.2 ± 21.1 | **69.2 ± 20.5** | 62.9 ± 21.1 | **66.4 ± 22.3** |
| Elastic transform | 70.5 ± 12.1 | 60.1 ± 12.0 | **69.9 ± 12.1** | 62.6 ± 11.9 | **69.3 ± 12.2** |
| Pixelate | 62.9 ± 22.0 | 58.1 ± 18.4 | **61.5 ± 22.7** | 58.4 ± 19.9 | **61.4 ± 21.8** |
| Jpeg compression | 60.2 ± 6.7 | 51.3 ± 6.2 | **59.8 ± 6.7** | 53.1 ± 6.2 | **59.0 ± 6.8** |
| Defocus blur | 76.3 ± 10.7 | 66.3 ± 13.3 | **75.9 ± 10.7** | 69.0 ± 12.1 | **75.3 ± 10.7** |
| Motion blur | 69.2 ± 8.6 | 57.7 ± 9.8 | **68.9 ± 8.6** | 60.9 ± 9.4 | **67.8 ± 8.6** |
| Zoom blur | 72.9 ± 6.1 | 61.5 ± 7.0 | **72.6 ± 6.1** | 64.5 ± 7.0 | **71.9 ± 6.1** |

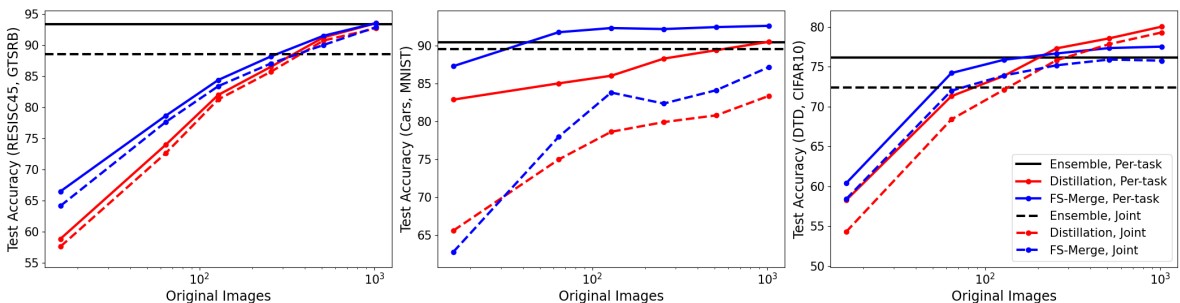

Figure 4: We used Ensemble, Distillation, and FS-Merge to merge pairs of models trained on RESISC45 and GTSRB (left), Cars and MNIST (center), DTD and CIFAR10 (right). We varied the number of original images per dataset and added augmentation images so the total number of images per dataset would be 1024. We present the per-task and joint accuracy.

tasks were merged in a low-data scenario of using only 16 original images and 800 augmented images per dataset. To study the impact of merging a larger number of models, Table 4 shows results for merging groups of four ViT-B-16 models. For additional results and hyperparameter details, refer to Appendices F.2 and E.2.

As can be seen, FS-Merge outperforms all other merging methods, and even surpasses ensembles in some cases. This holds for both per-task and joint accuracy across all settings. Additionally, it is evident that all local and simple methods (such as Average, SLERP, RegMean, and "Opt") completely fail to effectively merge ViTs in this challenging setting, resulting in a merged model that performs comparably to a random guess. Similar results were observed when merging larger ViTs (Appendix F.2) and ViTs trained with different pretraining strategies (Appendix F.3).

**Robustness evaluation.** In the next section, the robustness of merging methods under distribution shifts was evaluated. The ViT-B-16 model fine-tuned on CIFAR100 was merged with other ViT-B-16 models using KD and FS-Merge. The merged models were then tested on CIFAR100-C (Hendrycks & Dietterich, 2019), a dataset that introduces various corruptions at five severity levels to the CIFAR100 test set. For each corruption type, the mean and standard deviation across the severity levels were calculated (Table 5). The performance of the original ViT-B-16 model trained on CIFAR100 is also included as an upper bound for the merged models.

As demonstrated, FS-Merge consistently outperforms KD across corruption types and severity levels, achieving performance close to the original ViT-B-16 model trained on CIFAR100. This highlights its effectiveness in producing signficantly more robust merged models.

**Number of original training images.** We examine the impact of varying $|D|$, the number of images from the training datasets of the models to be merged ("original images"), which are used to create features. $|D|$ was varied from 16 to 1024. Augmented images were created to ensure the total number of images per dataset reached 1024, maintaining a consistent dataset size. Pairs of ViT-B-16 models were merged using Ensemble, KD and FS-Merge. The per-task and joint accuracies on the test set are presented in Figure 4. For additional details and experiments, see Appendix F.8.

As observed, Distillation underperforms with few original images, while FS-Merge excels. Increasing original images enhances both techniques, reducing the performance gap. With enough data, merging methods can sometimes surpass ensemble performance, which is often considered as a "gold-standard method" in merging and multitask articles.

### 3.3 Merging Text Transformers

In this series of experiments, we aimed to evaluate FS-Merge on a different modality: merging text transformers. We fine-tuned differently initialized BERT models on distinct classification task from the GLUE dataset (Wang et al., 2019). Then, we applied KD and FS-Merge to merge pairs of them, utilizing 200 data points

Table 6: Merging pairs of BERTs with 200 data points from the training set of each dataset. We report the test set per-task accuracy.

| Tasks | Original models | Distillation | FS-Merge |
|---|---|---|---|
| RTE, QQP | 79.49 | 65.00 | **68.88** |
| MNLI, MRPC | 85.73 | 76.87 | **78.29** |
| MRPC, QNLI | 89.02 | **80.67** | 79.33 |
| SST-2, RTE | 80.15 | 75.45 | **76.00** |
| MNLI, SST-2 | 88.39 | 82.81 | **83.24** |
| RTE, QNLI | 79.75 | 66.40 | **69.69** |
| QNLI, QQP | 91.01 | 82.40 | **84.75** |
| QQP, RTE | 79.49 | 73.30 | **75.62** |
| SST-2, QNLI | 91.68 | 82.92 | **84.39** |
| MRPC, RTE | 77.49 | **68.70** | 68.37 |
| QNLI, MNLI | 87.99 | 71.83 | **73.79** |
| QQP, SST-2 | 91.41 | 87.49 | **87.94** |
| MNLI, QQP | 87.7 | 78.74 | **79.22** |
| RTE, MRPC | 77.49 | 63.46 | **66.05** |

from each training set to create features. In Table 6, we report the per-task test accuracy of these experiments. We observe that FS-Merge outperforms KD, similar to the ViT case. For details and additional results, see Appendix F.7.

## 4  Discussion

**Merging Complexity.** FS-Merge and KD are more computationally intensive than standard merging methods such as Averaging and RegMean, which fail catastrophically in our setting. When merging two models, FS-Merge and KD have comparable resource usage. However, when merging multiple models, the resource usage gap between FS-Merge and KD becomes more significant. To address this, we propose FS-Merge seq., which is comparable to KD's resource use while outperforming it in terms of accuracy. Therefore, FS-Merge is recommended for optimal test accuracy, given sufficient computational resources and limited data. If resources are more limited, FS-Merge seq. should be the method of choice. Refer to Appendix C for the full details.

**Why does FS-Merge work well?** We attribute FS-Merge's superior performance to three key aspects. First, FS-Merge and KD offer greater expressiveness than traditional methods (Appendix H). Second, in contrast to the traditional methods, FS-Merge and KD use a global objective, which our experiments show is crucial in this challenging setting (see Appendix G.3). Lastly, we believe FS-Merge generalizes better than KD because it ha s an useful inductive bias that constrains the merged model to be a low-rank weighted average of the original models' weights (as determined by the Foldable SuperNet).

**Limitations.** FS-Merge requires a small unlabeled subset of the original training data, similarly to most previous merging methods. Additionally, FS-Merge, like KD, is more computationally intensive than standard merging methods; however, these methods completely failed in more challenging settings. Moreover, FS-Merge has fewer learnable parameters than KD, but they increase linearly with the number of models and hidden width; however, FS-Merge seq. solved the first issue. Lastly, one cannot naively merge two models of different depths using our method; we believe this could be solved in future work.

**Summary.** In this work, we address the challenging task of merging transformers from different initializations and tasks into a unified multitask model using a small subset of unlabeled data—a setting in which traditional methods fail. Our proposed FS-Merge, which employs a feature reconstruction approach to train a Foldable SuperNet, is simple, data-efficient, and can use more sophisticated merging rules compared to other baselines.

FS-Merge outperforms traditional methods and achieves SOTA results across various scenarios, model sizes, datasets and modalities.

**Impact Statement.** FS-Merge addresses key challenges in AI development. The ability to merge models from different initializations and tasks offers an efficient alternative to using an ensemble of these models. This approach enables greater resource efficiency and reduces the model's carbon footprint. Moreover, FS-Merge can alleviate privacy concerns. For instance, in cases where multiple users train models on private datasets, FS-Merge allows these models to be combined into a single multi-task model without accessing the full private datasets or any labels.

## Acknowledgements

The research of DS was Funded by the European Union (ERC, A-B-C-Deep, 101039436). Views and opinions expressed are however those of the author only and do not necessarily reflect those of the European Union or the European Research Council Executive Agency (ERCEA). Neither the European Union nor the granting authority can be held responsible for them. DS also acknowledges the support of the Schmidt Career Advancement Chair in AI. HM is the Robert J. Shillman Fellow, and is supported by the Israel Science Foundation through a personal grant (ISF 264/23) and an equipment grant (ISF 532/23).

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

# A    Related work

## A.1    Related work

**Mode Connectivity.**    A pair of models has mode connectivity when there exists a simple path between them in the loss landscape with low loss (Garipov et al., 2018; Draxler et al., 2018; Lubana et al., 2023). Frankle et al. (2020) demonstrated that this path can be linear (LMC) when the models originate from the same initialization. Tatro et al. (2020); Ainsworth et al. (2023); Entezari et al. (2022) showed that it is feasible to establish LMC between MLPs and CNNs from different initializations using permutations. Jordan et al. (2023) enhanced these solutions by addressing their variance collapse. Wang et al. (2020) used these permutations to match and then merge models within a federated learning framework. Imfeld et al. (2023); Verma & Elbayad (2024) used permutations to align two transformers with different initializations that were trained on the same task. Mirzadeh et al. (2021) leveraged LMC for multitask and continual learning. Yunis et al. (2022) expanded LMC to more than two models, discovering a high-dimensional convex hull of low loss. Zhou et al. (2023) introduced Layerwise Linear Feature Connectivity (LLFC), demanding that the features of the models be linearly connected. Singh & Jaggi (2020); Akash et al. (2022) addressed a similar challenge of neuron alignment using optimal transport (Knight, 2008), and Liu et al. (2022) generalized it as a graph-matching task. Navon et al. (2023) solved the neuron alignments problem by training an equivariant deep weight space network. It is important to note that previous works which merged models from different initializations using permutations primarily focused on merging pairs of models trained on the same task, with many of them concentrating on MLPs and CNNs. In contrast, our method is capable of merging larger groups of models, with different initializations and tasks, and is specifically designed to handle transformers.

**Model merging.**    Model merging technique (Goddard et al., 2024) has gained increasing interest in the past years, allowing the creation of stronger single-task and multi-task models. Akhlaghi & Sukhov (2018) showed that averaging the weights of multiple simple neural networks maintains their performance. Model soups (Wortsman et al., 2022) proposed averaging multiple models trained on the same task from identical initializations to enhance task accuracy. In addition, Weight averaging has been employed for various purposes, including improving optimization through checkpoint averaging (Izmailov et al., 2018), federated learning (McMahan et al., 2017), developing superior pre-trained models (Choshen et al., 2022; Don-Yehiya et al., 2022), improving Out-of-Distribution Generalization (Rame et al., 2022), achieving success in unseen tasks (Huang et al., 2023), and as augmentations for weight space networks (Shamsian et al., 2024). In many cases, spherical linear interpolation (SLERP) (Shoemake, 1985) is used to average the models' weights. Matena & Raffel (2022) utilized Fisher-Weighted Averaging to fuse multiple models from the same initializations but trained on diverse tasks, resulting in a multi-task model. RegMean (Jin et al., 2023) addressed a similar scenario and proposed a closed-form solution that solves a local linear regression problem for each linear layer in the model. Ilharco et al. (2023) defined task vector by subtracting the parameters of a fine-tuned model from those of the pre-trained model, and used it to fuse models fine-tuned from the same pre-trained model. Yadav et al. (2024); Ortiz-Jimenez et al. (2024); Yang et al. (2024); Akiba et al. (2024) analyzed and proposed more merging methods based on task vectos. Sung et al. (2023) fused pairs of models trained on different modalities. ZipIt (Stoica et al., 2024) merged models from various initializations and tasks, focusing on MLPs and CNNs, by averaging pairs of highly correlated neurons between and within the models (see Appendix A.2 for more details). In contrast, our work focuses on MLPs and transformers, and can use much more complicated merging schemes.

**Distillation.**    In Knowledge Distillation (KD), a small student model is trained to mimic the outputs of one or more larger teacher models (Ba & Caruana, 2014; Hinton et al., 2015; Stanton et al., 2021; Gou et al., 2021). Sau & Balasubramanian (2016) implemented a noise-based methodology to simulate learning a single task from multiple teachers. Tan et al. (2019); Khanuja et al. (2021) used several teachers, each translating between a specific language pair, to train a singular multilingual student. Clark et al. (2019); Park & Caragea (2023); Chelaramani et al. (2021) utilized multiple models and true labels to instruct a multi-task model. Pham et al. (2023) employed numerous quantized teachers trained on the same task to teach a quantized model, also leveraging true labels. Wu et al. (2021a) co-fine-tuned multiple teachers in downstream tasks with shared layers to instruct a student. Jacob et al. (2023) simultaneously trained multiple single-task models

with a single multi-task student to facilitate the student's optimization. Similar to our distillation baseline, Vongkulbhisal et al. (2019) utilized KD to unify knowledge from various models with different targets into a single model, relying solely on unlabeled data. Wu et al. (2021a); Zagoruyko & Komodakis (2017); Heo et al. (2019b;a) focused on using a single teacher's outputs and inner features to train a single model. Park & Kwak (2019); Liu et al. (2020) applied multiple teachers trained on the same task, their inner features, and the true labels to train a single-task student. Inspired these works, we tried to merge transformers using a loss function that takes into account the inner features. However, this approach proved ineffective for both our method and distillation. See Appendix G.5 for details. Wan et al. (2024) generates next-token probability distributions from multiple LLMs on a shared corpus, merges them into a single fused probability matrix, and uses it to transfer knowledge into a single LLM. Crucially, their setting does not involve models trained on different tasks. In contrast, our setting involves classification models with a shared architecture but trained on different tasks. Therefore, using a shared corpus to generate a fused probability matrix would not enable effective knowledge merging in our case.

**Adapter modules.** FS-Merge also resembles adapter-based methods (Houlsby et al., 2019; Chen et al., 2022; Zhang et al., 2024), which perform Parameter-Efficient Fine-Tuning (Hu et al., 2022; Zaken et al., 2021) of large models by injecting a small module ("adapters") into the original model. During fine-tuning, the original weights of the pre-trained model are frozen, and only the adapter weights are optimized. Similarly, FS-Merge injects new parameters into the model, which are the only ones optimized, though there are key differences. First, FS-Merge is used to merge the knowledge of multiple models, in contrast to adapter-based methods, which fine-tune a single model. Additionally, at the end of the training phase (or "merging"), the trainable parameters of FS-Merge are folded into the original weights, whereas in many adapter methods, the adapters remain in the fine-tuned models, adding complexity.

## A.2 Comparison to ZipIt

Our work strongly relates to ZipIt (Stoica et al., 2024), which also merges models from various initializations and tasks, with a focus on MLPs and CNNs. We will briefly explain the algorithm and its limitations. ZipIt uses the original training data to extract features from the models to be merged, $A$ and $B$. Then, for each layer $l$, it employs the features to identify pairs of highly correlated neurons using a greedy algorithm. Following this, it builds $M_l \in \mathbb{R}^{d_l \times 2 \cdot d_l}$, which is zero except for entries corresponding to a matched pair $(i, j)$ indexed by $p$, where $M_l[p, i] = M_l[p, j] = \frac{1}{2}$. The purpose of this matrix is to merge features by averaging pairs of highly correlated neurons. Then, $U_l = 2M_l^\top$ is used to reposition the merged features back to their original locations. After building $M_l$ and $U_l$, ZipIt proposes folding them with the original weights (Eq. 5) to create the weights of the merged model.

Like other current model merging methods (Ainsworth et al., 2023; Jin et al., 2023; Imfeld et al., 2023), ZipIt is efficient and focuses only on simple fusion schemes. It limit itself to pairing similar neurons, addressing only the permutation symmetries of neural networks (Hecht-Nielsen, 1990). Permutation symmetries mean that it is possible to swap any two neurons of a hidden layer in a neural network without altering its functionality. However, ZipIt falls short in handling the scale symmetries of neural networks (Neyshabur et al., 2015; Badrinarayanan et al., 2015; Phuong & Lampert, 2019), or in considering more complicated merging rules.

Probably due to these limitations, ZipIt underperforms on a large scale (such as ResNet-50 trained on datasets with 200 categories). Furthermore, this merging method is formulated as local problems, merging one layer at a time, and relies on heuristics to handle more complicated layers (such as batch normalization and skip connections). This makes it difficult to generalize to more complex architectures like transformers, where ZipIt struggles with self-attention and skip connection structures.

Our work aims to solve a similar problem, but adopts a more expressive approach without relying on heuristics. Moreover, the global version of our method allows the merging of any architecture by simply constructing a Foldable SuperNet that is suitable for it.

# B    Merging Vision Transformers with FS-Merge

## B.1    Foldable SuperNet for Vision Transformers

Assuming there are two Vision Transformers (ViTs) (Dosovitskiy et al., 2021), trained on two distinct tasks, $A$ and $B$, we aim to define a Foldable SuperNet that combines the original weights of the ViTs with new learnable parameters $M, U$. This structure is designed so that all skip connections and layer normalizations (Ba et al., 2016; Xiong et al., 2020) operate on the merged dimension, and that there is a linear layer before or after every $M$ or $U$ matrix, allowing them to be folded after training. It is important to highlight that layer norms possess significantly fewer learnable parameters compared to other layers in ViTs. Therefore, we can initiate their parameters from a good starting point (e.g., the parameters of the first model to be merged) and proceed to optimize the parameters as usual, similar to strategies employed in previous merging works (Jordan et al., 2023; Stoica et al., 2024).

The method described here for merging two models can be readily extended to any number of models. The parameters optimized at this stage are highlighted in red. The notation $\bar{X}$ represents the outputs of a layer norm, $\tilde{X}$ represents a feature reconstruction attempt after using the $U$ matrix, and $X^*$ represent the merged features.

### B.1.1    Pre-processing

First, each ViT creates patches from the input image and reshapes them into a series of vectors $I_{\text{proj}} \in \mathbb{R}^{T-1 \times d_{\text{in}}}$, using $W^{\text{in}} \in \mathbb{R}^{d_{\text{in}} \times d}$ to project these vectors into tokens. It then concatenates the $\text{CLS} \in \mathbb{R}^d$ token and adds $\text{emb} \in \mathbb{R}^{T \times d}$ which is the positional encoding. For example, for model $A$:

$$Z^A = \text{Concat}[I_{\text{proj}} W^{\text{in},A}, \text{CLS}^A] + \text{emb}^A .$$

The tokens from both models are then concatenated to form $(Z^A || Z^B) \in \mathbb{R}^{T \times 2 \cdot d}$, and the Foldable SuperNet merges them using a learned matrix $M_{\text{in}} \in \mathbb{R}^{2 \cdot d \times d}$ to produce $Z^* \in \mathbb{R}^{T \times d}$.

$$Z^* = (Z^A || Z^B) M_{\text{in}} .$$

Subsequently, and like the ViT, the Foldable SuperNet applies layer normalization, with parameters $\gamma^*_{-1}, \beta^*_{-1} \in \mathbb{R}^d$, to generate the input for the transformer. These parameters will be optimized.

$$\bar{X}^*_0 = \text{LN}_{\gamma^*_{-1}, \beta^*_{-1}}(Z^*)$$

### B.1.2    The attention block

The Foldable SuperNet of the attention block at layer $l$ (Figure 3) receives merged features $X^*_l \in \mathbb{R}^{T \times d}$ as inputs, and applies layer normalization. It's parameters $\gamma^*_l, \beta^*_l \in \mathbb{R}^d$ will be learned. Additionally, a learnable matrix $U_l \in \mathbb{R}^{d \times 2 \cdot d}$ is introduced to reconstruct the original features from the merged ones.

$$\tilde{X}^A_l || \tilde{X}^B_l = \text{LN}_{\gamma^*_l, \beta^*_l}(X^*_l) U_l .$$

After the layer normalization, the queries, keys, and values of each ViT are calculated. Then the Foldable SuperNet concatenates these components, from all heads and models, and merges them using a learnable matrix. Taking the queries as an example:

$$\tilde{Q}^A_l || \tilde{Q}^B_l = (\tilde{X}_l^A W^{Q,A}_l || \tilde{X}_l^B W^{Q,B}_l) \in \mathbb{R}^{T \times 2 \cdot d} ,$$

$$Q^*_l = (\tilde{Q}^A_l || \tilde{Q}^B_l) M^Q_l = (Q^*_{1,l}, ..., Q^*_{H,l}) .$$

The weights $W^{Q,A}_l, W^{Q,B}_l \in \mathbb{R}^{d \times d}$ generate queries from the embeddings. The matrix $M^Q_l \in \mathbb{R}^{2 \cdot d \times d}$ merges these features, which are then divided into $H$ heads, where each head $Q^*_{1,l}$ has dimensions $\mathbb{R}^{T \times \frac{d}{H}}$. Similarly, the keys and values are created by $W^K_l, W^V_l \in \mathbb{R}^{d \times d}$, and compressed using the matrices $M^K_l, M^V_l \in \mathbb{R}^{2 \cdot d \times d}$ respectively.

Following this, and similar to the ViT, the Foldable SuperNet executes multi-head attention with the merged queries, keys and values and concatenates the features from the heads. In our Foldable SuperNet, this step also includes adding $U_l^O \in \mathbb{R}^{d \times 2 \cdot d}$ to reconstruct the original multi-head attention outputs. Then each ViT utilizes $W_l^O \in \mathbb{R}^{d \times d}$ to aggregate those outputs. We also use $M_l^O \in \mathbb{R}^{2 \cdot d \times d}$ to compress it once more. This is followed by a skip connection.

$$Y_l^* = X_l^* + \text{Concat}_i[..., \text{softmax}\left(\frac{Q_{i,l}^* K_{i,l}^{* \top}}{\sqrt{d}}\right) V_{i,l}^*, ...] \, U_l^O \begin{pmatrix} W_l^{O,A} & 0 \\ 0 & W_l^{O,B} \end{pmatrix} M_l^O \, .$$

This process results in $Y_l^* \in \mathbb{R}^{T \times d}$, serving as the input for the subsequent MLP block at layer $l$.

### B.1.3   The Multi-Layer Perceptron block

The Foldable SuperNet of the $l$ MLP block receives $Y_l^* \in \mathbb{R}^{T \times d}$ as input, which are the merged features of the previous attention block. It learns the layer norm parameters of the MLP block $\alpha_l^*, \theta_l^* \in \mathbb{R}^d$, which, as usual, acts on the compressed dimension:

$$\bar{Y}_l^* = \text{LN}_{\alpha_l^*, \theta_l^*}(Y_l^*) \, .$$

After the layer norm, the ViT's MLP block applies a sequence of operations: a linear layer, an activation function, and another linear layer. Our Foldable SuperNet mimics this process and uses $M$ and $U$ matrices to both compress and reconstruct the features at each stage, akin to the approach described in Section 2.1. After these operations, a skip connection is applied on the compressed dimension.

$$X_{l+1}^* = Y_l^* + \sigma\left(\bar{Y}_l^* U_l^1 \begin{pmatrix} W_l^{1,A} & 0 \\ 0 & W_l^{1,B} \end{pmatrix} M_l^1\right) U_l^2 \begin{pmatrix} W_l^{2,A} & 0 \\ 0 & W_l^{2,B} \end{pmatrix} M_l^2 \, .$$

Where $U_l^1, U_l^2 \in \mathbb{R}^{d \times 2 \cdot d}$, $M_l^1, M_l^2 \in \mathbb{R}^{2 \cdot d \times d}$. $X_{l+1}^* \in \mathbb{R}^{T \times d}$ then serves as the input for the $l + 1$ attention block.

### B.2   Training the Foldable SuperNet

In the case of merging two ViTs $A$ and $B$, $D^A$ and $D^B$ are defined as small subsets of training data from tasks $A$ and task $B$ respectively.

Our objective is to define a global optimization problem for training the Foldable SuperNet. As in the case of linear layers, we aim to reconstruct the features from the last representation layer (just before the classification head) of the original ViTs. $I_{\text{img}}^k$ represents an input image from task $k$, and $f_L^k(I_{\text{img}}) \in \mathbb{R}^d$ represents the features from the last representation layer of the original model fine-tuned on task $k$, created from the input $I_{\text{img}}$. Observe that $f_L^k$ is the CLS token after being processed by the transformer and various post-processing stages that should also be merged (for instance, final layer normalization and a linear projection layer).

We will define the output of the Foldable SuperNet as $\tilde{f}_L(I_{\text{img}}) \in \mathbb{R}^{2 \cdot d}$, which is a reconstruction attempt for $f_L^A(I_{\text{img}}) || f_L^B(I_{\text{img}}) \in \mathbb{R}^{2 \cdot d}$. Also, $\tilde{f}_L(I_{\text{img}})[k]$ will note the reconstruction attempt for model $k$ features. Then the loss function will be:

$$L_{\text{out}} = \sum_k \mathbb{E}_{I_{\text{img}}^k \sim D^k} \left\| f_L^k(I_{\text{img}}^k) - \tilde{f}_L(I_{\text{img}}^k)[k] \right\|_2^2 \, .$$

This implies that for the input $I_{\text{img}}^k$ belonging to task $k$, we will only learn from the features of the model trained on this task, and not for example from the features that the model $j$ created $F^j(I_{\text{img}}^k)$. This loss differs from the one used in the MLP case, where each layer attempts to reconstruct the features that both models create from the input $I_{\text{img}}^k$, regardless of the task it belongs to. This method was adopted for both FS-Merge and KD when merging ViTs, as we found it performed better.

In the case of merging ViTs, this global approach worked much better than addressing a series of local problems for each block, as was done in the MLP case. For more details, please refer to Appendix G.3.

### B.3  Folding the Foldable SuperNet

Our next step after learning involves folding the Foldable SuperNet. This procedure aims to create a merged ViT that operates within the same dimensionality as the original models. The layer norm parameters acquired through our optimization process will be directly used in the merged model, as they already work in the merged dimension.

The folding operation (Eq. 5 ) follows the methodology outlined in Section 2. For instance, within the pre-processing block, the new merged projection weights and positional embeddings will be defined as follows:

$$W^{\text{in},*} = (W^{\text{in},A}||W^{\text{in},B})M_{\text{in}} \,,$$

$$\text{emb}^* = (\text{emb}^A||\text{emb}^B)M_{\text{in}} \,.$$

Also, taking the attention block at layer $l$ as an example, the merged query weights will be:

$$W^{Q,*} = U_l \begin{pmatrix} W_l^{Q,A} & 0 \\ 0 & W_l^{Q,B} \end{pmatrix} M_l^Q \,.$$

The other weights will be folded in a similar manner. Intuitively, this folding operation creates a merged model that, "under the hood", uses $U$ to reconstruct the original features from the previous layer, applies the original weights, and then uses $M$ to merge those features again, all with the same complexity as each of the original models.

### B.4  Merge tasks sequentially with FS-Merge seq.

Using the global version of FS-Merge on large models like transformers comes with a significant resource cost. As we have shown, the number of learnable parameters is smaller than in the distillation case, due to our modeling of the $M$ and $U$ matrices as a concatenation of diagonal matrices plus a low-rank matrix (Eq. 7). However, we still need to retain the frozen original weights of all models in memory, leading to increased memory and compute resource demands when merging multiple models.

To address this issue, we introduce FS-Merge Seq., which merges the models sequentially. For example, if we wish to merge $n$ models, we start by merging the first two models using the global FS-Merge (Eq. 4), with the features of these two original models as targets. The $M$ and $U$ matrices are still modeled as a concatenation of diagonal matrices plus a low-rank matrix, and we apply the "First" initialization (initialized from the first model).

After merging the first two models, we use global FS-Merge again to merge the resulting model (capable of solving the first two tasks) with the third model. In this phase, we use the features of all models seen so far (the first, second, and third models) as targets, and initialize from the merged model obtained in the previous step. This process is repeated, merging the previous merged model with a new original model at each step, until all $n$ models are merged. Note that In FS-Merge Seq., at each phase, we only merge and load the weights of two models (the previous merged model and a new original model), even though all features and models are utilized.

Experiments merging groups of four and five ViTs demonstrate that FS-Merge Seq. requires significantly less memory and compute resources, and merges models faster compared to regular FS-Merge. While this approach results in slightly lower accuracy compared to regular FS-Merge, FS-Merge Seq. still achieves better performance than distillation in most cases.

## C  Time and memory complexity analysis

For memory complexity analysis, let's consider the merging of $n$ fully connected layers with weights $W \in \mathbb{R}^{d \times d}$ to one layer. When using distillation to merge these layers, $d^2$ learnable parameters are required as we train a single weight matrix for all $n$ models.

In the FS-Merge case, we will have $2nd^2$ learnable parameters in the $M$ and $U$ matrices. Additionally, we must hold $nd^2$ frozen weights in memory, which comes with reduced cost compared to learnable parameters

Table 7: Measuring the total time and the number of optimized parameters, while merging a group of four ViT-B-16 with 100 original images and 1000 augmented images from each dataset. The per-task test accuracy is also reported. The merged models are the models fine-tuned on RESISC45, EuroSAT, CIFAR10, and MNIST.

| Method | Merging time | #Parameters Optimized | Accuracy |
|---|---|---|---|
| Average | $\sim$ 4 Seconds | 0 | 8.33 |
| SLERP | $\sim$ 4 Seconds | 0 | 8.69 |
| RegMean | $\sim$ 3 Minutes | 0 | 8.33 |
| Opt | $\sim$ 18 Minutes | 0 | 8.76 |
| Distillation | $\sim$ 1.9 Hours | 111M | 86.86 |
| FS-Merge diagonal | $\sim$ 3.2 Hours | 900K | 87.35 |
| FS-Merge low rank | $\sim$ 3.6 Hours | 60M | 91.63 |
| FS-Merge seq. | $\sim$ 2.2 Hours | 18M | 91.10 |

(as we do not need to compute gradients for these matrices, and the optimizer does not need to save their moments). Nevertheless, this 'vanilla' version of FS-Merge is much more memory-intensive compared to Distillation.

To mitigate this issue, we suggest two additional versions of FS-Merge. In the first one, we parameterize the $M$ and $U$ matrices as a concatenation of diagonal matrices plus a low-rank matrix with a rank of $r$ (Eq. 7), resulting in $2(nrd + nd + rd)$ learnable parameters. As demonstrated in the transformer case, small values of $r$ are sufficient to outperform distillation, which also results in fewer learnable parameters compared to distillation. However, we still need to retain $nd^2$ frozen weights in memory.

Our final and most efficient version is FS-Merge seq. (Appendix B.4), which merges a pair of models at each stage. Thus, we use the previous calculation with $n = 2$, which results in $6rd + 4d$ learnable parameters and $2d^2$ frozen weights in memory at each stage. FS-Merge seq. comes with a small cost to performance, but still outperform distillation (and see Section 3.2, Appendix F.2).

Table 7 presents the total time and the number of optimized parameters when merging a group of four ViT-B-16 models with 100 original images and 1000 augmented images from each dataset. FS-Merge seq. used with a low rank of 16. The per-task accuracy is also shown.

## D    Pre-training Vision Transformers

### D.1    Training Vision Transformers from scratch

We pre-trained ViT-B-16 and ViT-L-14 (Dosovitskiy et al., 2021; Steiner et al., 2022; Touvron et al., 2021) on ImageNet-1K (Deng et al., 2009). These models were initialized from distinct random seeds and exposed to training data in different orders. Following a setting similar to other merging works (Ilharco et al., 2023; Stoica et al., 2024), a frozen classification head derived from CLIP's (Radford et al., 2021) label embeddings was used, in order to make the outputs space of the ViTs similar. Training and merging ViTs with learned classification heads are left for future research. It is important to mention that the classification heads are not being merged.

We pre-trained the ViTs following common practices (Touvron et al., 2021), such as augmentations, MixUp (Zhang et al., 2018), distillation using resnet152 (He et al., 2016), cross-entropy loss, and a cosine scheduler with a single cycle and a warmup.

### D.2    Datasets and fine-tuned Vision Transformers

After obtaining different pre-trained ViTs, we fine-tuned each ViT on a different downstream task. Following the approach outlined in (Ilharco et al., 2023), we fine-tuned on a range of datasets including Cars (Krause

Table 8: Dataset details and the test accuracy of the fine-tuned ViT-B-16 and ViT-L-14.

| Dataset | Train size | Val size | Test size | #Classes | ViT-B-16 test acc. | ViT-L-14 test acc. |
|---|---|---|---|---|---|---|
| EuroSAT | 18,700 | 3,300 | 5,00 | 10 | 99.06 | 98.10 |
| GTSRB | 22,644 | 3,994 | 12,630 | 43 | 98.32 | 97.88 |
| Cars | 6,923 | 1,221 | 8,041 | 196 | 86.12 | 93.32 |
| CIFAR-10 | 42,500 | 7,500 | 10,000 | 10 | 97.33 | 98.9 |
| CIFAR-100 | 42,500 | 7,500 | 10,000 | 100 | 85.61 | 92.74 |
| DTD | 1,880 | 1,880 | 1,880 | 47 | 64.04 | 64.57 |
| MNIST | 51,000 | 9,000 | 10,000 | 10 | 99.66 | 99.68 |
| RESISC45 | 18,900 | 6,300 | 6,300 | 45 | 93.46 | 93.95 |
| SVHN | 62,269 | 10,988 | 26,032 | 10 | 96.78 | 96.63 |

et al., 2013), DTD (Cimpoi et al., 2014), EuroSAT (Helber et al., 2019), GTSRB (Stallkamp et al., 2011), MNIST (LeCun, 1998), RESISC45 (Cheng et al., 2017), SVHN (Netzer et al., 2011), CIFAR10, and CIFAR100 (Krizhevsky et al., 2009). We used the existing datasets' training set, validation set, and test set. If there wasn't a validation set, one was created by using 15% of the training set.

All models were fine-tuned with a batch size of 256, a learning rate of $1e^{-5}$, cross-entropy loss, and a cosine scheduler using a single cycle with a warm-up phase. As done in previous works (Ilharco et al., 2023; Stoica et al., 2024), the classification head was frozen and obtained from CLIP embeddings (Radford et al., 2021) of the label names. In Table 8, we present the dataset details and the test accuracy of the fine-tuned models on their respective tasks. The models' accuracies on tasks they were not fine-tuned for are as good as a random guess.

# E  Merging experiments details and hyperparameters

## E.1  Merging Multi-Layer Perceptrons

We divided the MNIST dataset (LeCun, 1998) into two subsets: the first containing labels 0 to 4 and the second containing labels 5 to 9. Each subset was further split into training, validation, and testing sets, with the validation set comprising 10% of the original training set. A similar approach was adopted for the Fashion-MNIST dataset (Xiao et al., 2017).

We trained MLPs on each of the MNIST subsets, utilizing SGD with a learning rate of 0.01, a batch size of 10, and cross-entropy loss. The training duration was set to 1 epoch. However, if the number of hidden layers exceeded four, or the hidden dimension surpassed 256, the training was extended to 2 epochs. For the Fashion MNIST dataset (Xiao et al., 2017), the same hyperparameters were used, with adjustments made only to the number of epochs, increased to 10, and the learning rate, reduced to 0.001.

**Data.** All the merging methods, with the exception of simple weight averaging, need features generated from the MLPs. Those were created by using unlabeled data from the training sets. 64 images from each split dataset were utilized, resulting in total of 128 images. Additionally, we normalized the target features of the two models to the same scale in FS-Merge and in distillation, because it led to an improvement in accuracy. Note that this will not hurt the performance of the merged model during inference, as the scale of the last layer's features does not change the prediction.

**Hyperparameters for the merging methods.** The hyperparameters for the merging methods were determined separately for MNIST and Fashion MNIST. We chose the hyperparameters that maximize the per-task accuracy on the validation set when merging two MLPs with two hidden layers and a hidden width of 128. We then used those hyperparameters for merging MLPs with different depths and widths.

We will now outline the hyperparameters grid used for the hyperparameter search in each merging method. We used GD optimizer in FS-Merge and Distillation (so the batch size is 128, the whole data). The step learning rate scheduler employs two learning rate drops with $\gamma = 0.9$, whereas the Cosine scheduler utilizes a single cycle with a warmup length of 20 epochs.

FS-Merge.

- initialization type: "random"

- num epochs: [1k, 5k, 10k, **15k** (MNIST), **20k** (Fashion MNIST), 25k]

- learning rate: [0.3, **0.1**, 0.03, 0.01, 0.003]

- momentum: [**0.9**, 0.8]

- scheduler: [**step lr**, cosine]

FS-Merge global.

- initialization type: ["First", **"Average"** (the average of the original models), "random"]

- num epochs: [200, 400, **1k** (MNIST), **1.5k** (Fashion MNIST), 5k, 10k, 15k]

- learning rate: [0.3, **0.1**, 0.03, 0.01, 0.003]

- momentum: [0.9, **0.8**]

- scheduler: [step lr, **cosine**]

FS-Merge global, ZipIt init.

- initialization type: [**"ZipIt"**]

- num epochs: [200, 400, 1k, 1.5k, 5k, **10k**, 15k]

- learning rate: [0.3, **0.1**, 0.03, 0.01, 0.003]

- momentum: [**0.9**, 0.8]

- scheduler: [step lr, **cosine**]

Distillation.

- initialization type: ["First", **"Average"** (the average of the original models), "random"]

- num epochs: [200, 400, **1k** (MNIST), **1.5k** (Fashion MNIST), 2k, 5k]

- learning rate: [0.3, 0.1, **0.03**, 0.01, 0.003]

- momentum: [0.9, **0.8**]

- scheduler: [step lr, **cosine**]

RegMean.

- $\alpha$: [1.0, **0.9** (MNIST), **0.8** (Fashion MNIST), 0.7, 0.6, 0.5, 0.4, 0.3, 0.2, 0.1]

Table 9: The different experimental settings. "#Original Images" refers to the number of original images taken from the training set per dataset. "#Augmented Images" refers to the number of augmented images created per dataset. The fine-tuned models refer to the models used for the hyperparameter search. When C10 = CIFAR10, G = GTSRB, R = RESISC45, S = SVHN.

| Setting | What is merged | #Original Images | #Augmented Images | Total #Images | fine-tuned models |
|---|---|---|---|---|---|
| a | 2 ViT-B-16 | 16 | 800 | 1,632 | R, C10 |
| b | 4 ViT-B-16 | 100 | 1000 | 4,400 | R, C10, S, G |
| c | 2 ViT-L-14 | 100 | 1000 | 2,200 | R, C10 |

### E.2 Merging Vision Transformers

**Data and augmentations.** We took a set of images from the training set and expanded it using augmentations. Specifically, these augmentations were used: Random Crop; Random Horizontal Flip; Random choice between grayscale, Solarization, and GaussianBlur; and MixUp (Zhang et al., 2018). Then this dataset is used to create features, where features of the model fine-tuned on task *A* were generated only from images of this task (and their augmentations). For efficiency and reproducibility, we saved these features and then used them for all our merging methods. In the case of FS-Merge and distillation, a single epoch means using the entire features dataset once, including the features created by augmentations.

**Hyperparameter.** Three types of hyperparameter experiments were conducted: merging pairs of ViT-B-16 in a low-data scenario, merging groups of four ViT-B-16, and merging pairs of ViT-L-14. For each experimental setting, a specific group of fine-tuned models was selected and a hyperparameter search was performed. The hyperparameters that maximized the per-task validation accuracy for this group were chosen and then applied when merging other model groups in this setting. The settings can be seen in Table 9.

For all methods that require training (FS-Merge and distillation), a batch size of 128 was used, along with a cosine scheduler that utilized a single cycle with a warmup phase, an ADAMW optimizer with a weight decay of 0.001, and initialization from the first model ("First"). Similar to the MLP experiments, for the distillation baseline, the target features were scaled to an L2 norm of 0.5. However, in this specific setting, this scaling proved unhelpful for FS-Merge and was therefore not utilized. The hyperparameter grid used for the hyperparameter search will now be outlined.

FS-Merge, concatenation of diagonal matrices, without a low rank matrix.

- epochs: [30, **100** (c), **200** (a, b), 300, 400]

- lr: [0.1, **0.01** (c), **0.001** (a, b), 0.0001]

FS-Merge, concatenation of diagonal matrices + low rank matrix.

- epochs: [30, 100, **200**, 300, 400]

- lr: [0.1, 0.01, 0.001, **0.0001**, 0.00001]

FS-Merge seq. (Appendix B.4). "epochs" refers to the number of epochs used for all iterations except the last one. "last iteration epochs" denotes the number of epochs applied in the last iteration, which involves merging the final model with the model obtained from the previous iteration.

- epochs: [10, **50**, 100]

- last iteration epochs: [50, 100, **200**, 300]

- lr: [**0.0001**]

Distillation.

- epochs: [30, **100** (a, c), 200, **300** (b), 400]

- lr: [0.1, 0.01, 0.001, **0.0001**, 0.00001]

RegMean (Jin et al., 2023). $\alpha$ is a factor which decrease the non-diagonal items of the inner product matrices in the RegMean solution.

- $\alpha$: [1.0, **0.9** (a, c), 0.8, **0.7** (b), 0.6, 0.5, 0.4, 0.3, 0.2, 0.1]

"Opt" (Imfeld et al., 2023). The hyperparameters include the filter type (which determines the feature tokens used by the optimal transport solver) and $\lambda$, a regularization term for the solver (Sinkhorn-Knapp algorithm). Lower values of $\lambda$ result in harder alignment.

- filter: [Only CLS, **Full** (c), Window 2, Window 4, Window 6, **Window 8** (a, b), Window 10, Window 14]

- $\lambda$: [0, **0.08** (a,c), **0.2** (b), 0.5]

## F   Additional results

### F.1   Merging Multi-Layer Perceptrons

Additional results were obtained by merging pairs of MLPs, using 64 images from each task to create features. Each experiment was replicated five times with different random seeds. Note that we do not merge the last linear layer (the classification head).

**FS-Merge.** Our method was tested in five variants. FS-Merge is the local version (Eq. 3), which trains a Foldable SuperNet for each layer $l$ independently, using the original models' pre-activation features as inputs $z_l^A, z_l^B$. FS-Merge global is the global version of our method (Eq. 4), training a Foldable SuperNet for all the layers together to reconstruct the features of the final representation layer $f_L^A, f_L^B$. For both of these versions, we also tested a variant where we initialized the Foldable SuperNet's $M$ and $U$ with the solutions of ZipIt (Stoica et al., 2024), and then continued to optimize them (FS-Merge ZipIt init and FS-Merge global ZipIt init). FS-Merge no cross compresses each of the two MLPs ($A$ and $B$) individually to half of their widths using a local FS-Merge and then concatenates those two compressed models. This means that neurons between these two models cannot be merged.

We also tried a local FS-Merge version where the $l$-th Foldable SuperNet layer uses the reconstructed features from the previous layer $\tilde{z}_l^A, \tilde{z}_l^B$ as inputs, but it achieved the same accuracy as the regular local FS-Merge.

Table 10 presents the per-task accuracy for merging MLPs trained on half of the MNIST dataset (LeCun, 1998), with varying hidden width. Table 11 and Table 12 present the joint accuracy for fusing MLPs trained on half of the MNIST dataset, with variations in depth or hidden width, respectively. Similarly, Table 13 and Table 14 present the per-task accuracy for merging MLPs trained on half of the Fashion MNIST dataset (Xiao et al., 2017), again varying by depth or hidden width. The joint accuracy for these models are detailed in Table 15.

As shown in the experiments, our method, especially when using ZipIt initialization, demonstrates SOTA results across all settings and outperforms ensembles in some cases. It also appears that FS-Merge achieves better per-task accuracy, while global FS-Merge achieves better joint accuracy. Furthermore, ZipIt stands out as a strong baseline.

### F.2   Merging Vision Transformers

For this section, C = Cars, C10 = CIFAR10, C100 = CIFAR100, D = DTD, E = EuroSAT, G = GTSRB, M = MNIST, R = RESISC45, S = SVHN.

Table 10: Merging pairs of MLPs, each initialized differently and trained on distinct halves of the **MNIST** dataset. These MLPs have a single hidden layer, with the **Hidden width varying** from 16 to 1024. Each experiment was replicated five times. We present the average **per task accuracy** on the test set, along with the standard deviation.

| Merge Method | Hidden width | | | | |
|---|---|---|---|---|---|
| | 16 | 64 | 128 | 512 | 1024 |
| Original Models | 96.34 ± 0.3 | 96.77 ± 0.1 | 96.8 ± 0.1 | 97.8 ± 0.1 | 98.0 ± 0.1 |
| Ensemble | 84.47 ± 4.3 | 93.19 ± 1.8 | 94.0 ± 0.6 | 96.4 ± 0.3 | 96.5 ± 0.3 |
| average | 83.88 ± 3.9 | 93.05 ± 1.7 | 94.1 ± 0.4 | 96.6 ± 0.1 | 96.8 ± 0.1 |
| RegMean | 88.12 ± 3.6 | 95.06 ± 1.2 | 95.2 ± 0.2 | 96.7 ± 0.1 | 96.9 ± 0.2 |
| ZipIt | 91.84 ± 1.9 | 96.06 ± 0.12 | 96.3 ± 0.2 | **97.6 ± 0.1** | 97.7 ± 0.1 |
| Distillation | 82.69 ± 4.7 | 91.41 ± 2.5 | 93.0 ± 0.7 | 95.7 ± 0.4 | 96.0 ± 0.3 |
| FS-M | 76.7 ± 29.0 | 95.87 ± 0.1 | 95.8 ± 0.2 | 96.1 ± 0.2 | 95.9 ± 0.2 |
| FS-M, ZipIt init | **95.32 ± 0.4** | **96.50 ± 0.1** | **96.6 ± 0.1** | 97.5 ± 0.1 | **97.8 ± 0.1** |
| FS-M no cross | 63.8 ± 11.2 | 96.18 ± 0.1 | 96.1 ± 0.1 | 96.6 ± 0.1 | 96.4 ± 0.1 |
| FS-M global | 11.67 ± 4.2 | 95.66 ± 0.3 | 95.7 ± 0.1 | 96.3 ± 0.2 | 96.4 ± 0.2 |
| FS-M global, ZipIt init | 11.84 ± 4.6 | 96.44 ± 0.1 | **96.6 ± 0.1** | **97.6 ± 0.1** | **97.8 ± 0.1** |

Table 11: Merging pairs of MLPs, each initialized differently and trained on distinct halves of the **MNIST** dataset. These MLPs contain 128 neurons per hidden layer, with the **number of hidden layers varying** from 1 to 6. Each experiment was replicated five times. We present the average **joint accuracy** on the test set, along with the standard deviation.

| Merge Method | Number of Hidden Layers | | | | |
|---|---|---|---|---|---|
| | 1 | 2 | 3 | 4 | 6 |
| Ensemble | 80.20 ± 1.7 | 83.25 ± 3.3 | 84.6 ± 2.1 | 83.9 ± 1.9 | 83.1 ± 1.5 |
| average | 78.96 ± 0.7 | 69.83 ± 6.2 | 57.4 ± 5.2 | 38.2 ± 10.2 | 11.5 ± 2.9 |
| RegMean | 82.81 ± 1.5 | 80.38 ± 4.6 | 75.3 ± 2.8 | 74.7 ± 2.2 | 63.0 ± 7.0 |
| ZipIt | 85.33 ± 1.1 | 84.49 ± 1.9 | 81.5 ± 1.4 | 80.0 ± 1.4 | 80.2 ± 3.2 |
| Distillation | 77.32 ± 1.5 | 79.2 ± 3.96 | 79.6 ± 2.1 | 78.2 ± 3.6 | 74.4 ± 2.0 |
| FS-M | 84.44 ± 0.6 | 84.92 ± 1.0 | 82.7 ± 2.2 | 79.5 ± 3.1 | 79.1 ± 3.1 |
| FS-M, ZipIt init | **86.18 ± 0.6** | **86.57 ± 1.4** | 83.9 ± 3.1 | 79.7 ± 3.8 | 80.9 ± 3.6 |
| FS-M no cross | 85.39 ± 0.6 | 85.88 ± 1.3 | 86.0 ± 1.0 | 84.6 ± 1.1 | 83.8 ± 0.4 |
| FS-M global | 83.67 ± 0.7 | 84.12 ± 1.1 | 83.9 ± 1.0 | 82.8 ± 1.1 | 81.6 ± 0.7 |
| FS-M global, ZipIt init | **86.18 ± 0.7** | 86.40 ± 1.2 | **86.3 ± 1.2** | **84.7 ± 1.3** | **84.1 ± 0.9** |

Table 12: Merging pairs of MLPs, each initialized differently and trained on distinct halves of the **MNIST** dataset. These MLPs have a single hidden layer, with the **hidden width** varying from 16 to 1024. Each experiment was replicated five times. We present the average **joint accuracy** on the test set, along with the standard deviation.

| Merge Method | Hidden width | | | | |
|---|---|---|---|---|---|
| | 16 | 64 | 128 | 512 | 1024 |
| Ensemble | 58.86 ± 7.6 | 77.60 ± 5.5 | 82.1 ± 1.8 | 85.7 ± 0.5 | 87.0 ± 0.7 |
| average | 61.65 ± 8.0 | 77.16 ± 4.1 | 81.0 ± 2.2 | 84.4 ± 1.4 | 86.2 ± 0.6 |
| RegMean | 66.73 ± 7.7 | 81.54 ± 3.2 | 84.0 ± 1.6 | 85.2 ± 1.1 | 87.1 ± 1.0 |
| ZipIt | 66.89 ± 2.9 | 85.21 ± 1.0 | 83.9 ± 1.8 | 87.7 ± 1.5 | 88.0 ± 1.5 |
| Distillation | 57.15 ± 7.8 | 74.98 ± 4.9 | 79.6 ± 1.8 | 83.4 ± 0.6 | 85.0 ± 0.6 |
| FS-M | 59.1 ± 26.2 | 84.30 ± 1.0 | 83.9 ± 1.0 | 85.6 ± 0.9 | 85.5 ± 0.5 |
| FS-M, ZipIt init | **77.52 ± 4.2** | 85.91 ± 1.1 | **86.3 ± 0.7** | 88.5 ± 1.0 | 89.3 ± 0.5 |
| FS-M no cross | 54.00 ± 9.3 | 85.34 ± 1.5 | 85.0 ± 1.2 | 86.5 ± 0.8 | 86.5 ± 0.4 |
| FS-M global | 9.46 ± 0.1 | 83.72 ± 1.6 | 83.5 ± 1.1 | 85.0 ± 1.0 | 85.7 ± 0.5 |
| FS-M global, ZipIt init | 9.63 ± 0.1 | **85.99 ± 1.3** | 86.1 ± 0.9 | **88.6 ± 0.9** | **89.4 ± 0.4** |

Table 13: Merging pairs of MLPs, each initialized differently and trained on distinct halves of the **Fashion MNIST** dataset. These MLPs contain 128 neurons per hidden layer, with the **hidden depth** varying from 1 to 6. Each experiment was replicated five times. We present the average **per task accuracy** on the test set, along with the standard deviation.

| Model | Number of Hidden Layers | | | | |
|---|---|---|---|---|---|
| | 1 | 2 | 3 | 4 | 6 |
| Original Models | 90.45 ± 0.14 | 90.53 ± 0.1 | 90.4 ± 0.1 | 89.9 ± 0.1 | 83.0 ± 2.2 |
| Ensemble | 87.31 ± 1.93 | 88.68 ± 0.6 | 86.2 ± 1.9 | 86.9 ± 2.0 | 77.7 ± 4.0 |
| average | 86.04 ± 2.40 | 78.20 ± 5.1 | 58.9 ± 10.3 | 47.8 ± 5.4 | 22.4 ± 3.0 |
| RegMean | 89.29 ± 0.49 | 86.29 ± 0.2 | 81.3 ± 3.3 | 76.7 ± 4.4 | 64.1 ± 5.5 |
| ZipIt | 89.24 ± 0.44 | 87.40 ± 0.5 | 85.4 ± 2.9 | 83.2 ± 1.4 | 70.2 ± 8.5 |
| Distillation | 86.94 ± 1.49 | 87.84 ± 0.8 | 83.0 ± 1.6 | 83.2 ± 2.4 | 69.2 ± 3.9 |
| FS-M | 89.86 ± 0.13 | 89.80 ± 0.1 | 89.1 ± 0.2 | 88.4 ± 0.5 | 73.1 ± 4.1 |
| FS-M, ZipIt init | **90.20 ± 0.12** | **90.28 ± 0.1** | **90.0 ± 0.1** | **89.7 ± 0.2** | 80.0 ± 2.0 |
| FS-M no cross | 90.03 ± 0.13 | 90.20 ± 0.1 | 89.9 ± 0.1 | 89.3 ± 0.2 | **82.4 ± 2.2** |
| FS-M global | 89.85 ± 0.15 | 89.81 ± 0.2 | 89.3 ± 0.1 | 88.5 ± 0.2 | 78.5 ± 2.4 |
| FS-M global, ZipIt init | 90.04 ± 0.08 | 89.95 ± 0.1 | 89.5 ± 0.2 | 88.4 ± 0.2 | 62.5 ± 2.4 |

Table 14: Merging pairs of MLPs, each initialized differently and trained on distinct halves of the **Fashion MNIST** dataset. These MLPs have a single hidden layer, with the **hidden width** varying from 16 to 1024. Each experiment was replicated five times. We present the average **per task accuracy** on the test set, along with the standard deviation.

| Merge Method | hidden width | | | | |
|---|---|---|---|---|---|
| | 16 | 64 | 128 | 512 | 1024 |
| Original Models | 90.31 ± 0.06 | 90.43 ± 0.1 | 90.4 ± 0.1 | 90.6 ± 0.1 | 90.7 ± 0.1 |
| Ensemble | 72.71 ± 4.72 | 85.21 ± 3.3 | 86.9 ± 1.0 | 88.9 ± 0.4 | 87.0 ± 1.0 |
| average | 70.33 ± 6.56 | 83.90 ± 4.2 | 86.2 ± 1.4 | 88.5 ± 0.4 | 86.8 ± 0.8 |
| RegMean | 74.90 ± 8.31 | 87.33 ± 2.5 | 88.7 ± 1.5 | 89.1 ± 0.3 | 87.1 ± 1.3 |
| ZipIt | 77.39 ± 7.60 | 88.73 ± 0.7 | 89.4 ± 0.6 | 89.8 ± 0.2 | 88.7 ± 0.2 |
| Distillation | 74.06 ± 5.12 | 85.49 ± 2.9 | 86.7 ± 1.3 | 88.4 ± 0.6 | 86.5 ± 1.0 |
| FS-M | 89.06 ± 0.64 | 89.90 ± 0.1 | 89.8 ± 0.1 | 89.5 ± 0.1 | 89.1 ± 0.1 |
| FS-M, ZipIt init | **90.10 ± 0.19** | **90.25 ± 0.1** | **90.2 ± 0.1** | **90.3 ± 0.1** | **90.3 ± 0.1** |
| FS-M no cross | 49.2 ± 12.63 | 90.09 ± 0.1 | 91.1 ± 0.1 | 89.8 ± 0.1 | 89.5 ± 0.1 |
| FS-M global | 13.98 ± 4.88 | 89.87 ± 0.1 | 89.9 ± 0.1 | 89.6 ± 0.1 | 89.4 ± 0.1 |
| FS-M global, ZipIt init | 11.98 ± 3.97 | 89.98 ± 0.1 | 90.1 ± 0.1 | 90.2 ± 0.1 | 90.1 ± 0.1 |

Table 15: Merging pairs of MLPs, each initialized differently and trained on distinct halves of the **Fashion MNIST** dataset. These MLPs contain 128 neurons per hidden layer, with the **hidden depth** varying from 1 to 6. Each experiment was replicated five times. We present the average **joint accuracy** on the test set, along with the standard deviation.

| Model | Number of Hidden Layers | | | | |
|---|---|---|---|---|---|
| | 1 | 2 | 3 | 4 | 6 |
| Ensemble | 55.94 ± 4.14 | 56.28 ± 2.0 | 57.7 ± 2.9 | 56.9 ± 2.2 | 53.0 ± 0.9 |
| average | 54.71 ± 4.04 | 42.76 ± 6.4 | 33.6 ± 5.7 | 21.8 ± 1.8 | 8.9 ± 2.0 |
| RegMean | 56.94 ± 1.24 | 54.78 ± 2.1 | 54.7 ± 4.3 | 50.0 ± 4.5 | 51.6 ± 5.1 |
| ZipIt | 59.26 ± 1.66 | **60.01 ± 1.5** | **60.9 ± 2.4** | 59.6 ± 2.5 | 48.3 ± 7.3 |
| Distillation | 55.70 ± 2.49 | 54.33 ± 2.5 | 54.0 ± 3.9 | 52.6 ± 4.0 | 45.5 ± 3.9 |
| FS-M | 54.40 ± 0.29 | 49.91 ± 1.1 | 49.1 ± 1.5 | 47.3 ± 1.6 | 42.2 ± 2.3 |
| FS-M, ZipIt init | 54.13 ± 0.28 | 48.71 ± 1.0 | 48.1 ± 1.1 | 47.3 ± 1.2 | 44.3 ± 1.0 |
| FS-M no cross | 56.81 ± 0.23 | 56.19 ± 0.9 | 56.4 ± 1.3 | 56.7 ± 1.0 | **53.0 ± 0.8** |
| FS-M global | 56.63 ± 0.27 | 55.64 ± 0.9 | 55.8 ± 1.4 | 56.1 ± 0.9 | 51.9 ± 0.9 |
| FS-M global, ZipIt init | **60.46 ± 0.94** | 59.31 ± 0.3 | 60.1 ± 0.5 | **61.7 ± 2.3** | 40.5 ± 3.4 |

Table 16: Merging pairs of ViT-B-16 with 16 original images from the training set and 800 augmented images from each dataset. We report the per-task and joint accuracy on the test set.

| Merging Methods | EuroSAT, CIFAR100 | | Cars, MNIST | | RESISC45, CIFAR10 | |
|---|---|---|---|---|---|---|
| | Per-task | Joint | Per-task | Joint | Per-task | Joint |
| Original models | 92.33 | - | 92.89 | - | 95.39 | - |
| Ensemble | 90.25 | 83.72 | 90.44 | 89.63 | 92.17 | 86.06 |
| Average | 5.635 | 1.94 | 4.96 | 4.87 | 5.61 | 1.66 |
| SLERP | 5.88 | 2.87 | 7.12 | 4.34 | 4.82 | 1.76 |
| RegMean | 4.45 | 0.95 | 5.18 | 0.27 | 8.45 | 5.44 |
| Opt | 5.60 | 0.90 | 5.37 | 5.16 | 6.32 | 5.32 |
| Distillation | **72.02** | **66.34** | 80.14 | 63.28 | 65.47 | 59.64 |
| FS-M diagonal | 69.53 | 63.98 | 85.88 | 73.10 | 72.11 | 65.68 |
| FS-M low rank | 70.86 | 65.23 | **87.92** | **76.24** | **76.35** | **70.96** |

**Baselines.** Our goal was to merge Vision Transformers (ViTs) that were initialized differently and trained on various tasks. We compared our method against "average" (Wortsman et al., 2022), a simple weight averaging technique; "SLERP" (Shoemake, 1985), spherical linear interpolation; RegMean (Jin et al., 2023), which offers a closed-form solution by solving a linear regression problem for each linear layer; "Opt" (Imfeld et al., 2023), which uses optimal transport (Knight, 2008) to align transformers, and can be viewed as a generalization of neuron alignment methods (Ainsworth et al., 2023) because it can find soft alignments as well as hard ones; and a distillation baseline (Hinton et al., 2015), which trains a single ViT to mimic the features of the last representation layer of the original models.

**FS-Merge.** We examined three versions of our method: FS-Merge low rank, where the $M$ and $U$ matrices were parameterized as a concatenation of diagonal matrices, plus a low rank matrix; FS-Merge diagonal, which is as "FS-Merge low rank" but with a low rank of 0; in the experiments of merging groups of four and five ViTs, we also used FS-Merge seq. (Appendix B.4), with a low rank of 16.

Additional results of the experiment involving the merging of pairs of ViT-B-16 can be seen in Table 16. This experiment examined our ability to merge models with a very low number of only 16 original images per dataset. An additional 800 images per dataset were created using augmentations. FS-merge low rank was used with a low rank of 12.

Table 17 presents experiments on merging groups of three ViT-B-16 models, each fine-tuned on different tasks, using 100 original images per dataset and 1000 augmented images per dataset. The FS-M low-rank method was applied with a rank of 28. All merging methods were used with the hyperparameters chosen for the experiment involving the merging of groups of four ViT-B-16 models.

Additional results of the experiment involving the merging of groups of four ViT-B-16 can be seen in Table 18 and Table 19. FS-merge low rank was used with a low rank of 32. In Table 20 we merged groups of five ViT-B-16, where FS-merge low rank was used with a low rank of 32. Note that solving five different tasks via merging is an extremely difficult challenge, and even ensemble struggles with it.

Table 21, Table 22 and Table 23 present additional results from the experiment involving the merging of ViT-L-14 pairs. This experiment investigated how the merging methods scale to larger models, with FS-Merge Low Rank applied using a rank of 32.

### F.3 Merging Vision Transformers with different pre-trained strategies

The next section investigates the ability of merging methods to merge vision transformers that used different pre-training strategies. ViT-B-16 models were pre-trained from scratch using supervised training on the ImageNet-1k dataset (Deng et al., 2009) (the pre-training strategy used throughout the rest of this work). These models were then fine-tuned on RESISC45 (Cheng et al., 2017), DTD (Cimpoi et al., 2014), SVHN

Table 17: Merging groups of 3 ViT-B-16 with 100 original images from the training set and 1000 augmented images from each dataset (resulting in a total of 300 original images and 3,000 augmented images). We report the per-task and joint accuracy on the test set. "Parameters Optimized" refers to the number of learnable parameters. We will denote: C = Cars, C10 = CIFAR10, C100 = CIFAR100, D = DTD, E = EuroSAT, G = GTSRB, M = MNIST, R = RESISC45, S = SVHN.

| Merging Methods | R, C100, S | | C100, C, M | | D, M, E | | #Parameters Optimized |
|---|---|---|---|---|---|---|---|
| | Per-task | Joint | Per-task | Joint | Per-task | Joint | |
| Original models | 91.95 | - | 90.46 | - | 87.58 | - | - |
| Ensemble | 84.28 | 48.49 | 83.92 | 61.67 | 82.71 | 63.02 | - |
| Average | 4.02 | 0.34 | 3.77 | 0.25 | 7.76 | 3.39 | 0 |
| SLERP | 3.88 | 0.69 | 3.69 | 0.08 | 8.84 | 5.11 | 0 |
| RegMean | 3.81 | 0.83 | 4.76 | 0.20 | 7.17 | 0.71 | 0 |
| Opt | 3.95 | 0.52 | 4.01 | 0.13 | 7.22 | 2.36 | 0 |
| Distillation | 65.03 | 52.36 | 59.79 | **49.64** | 79.04 | 76.76 | 111M |
| FS-M, diagonal | 63.77 | 45.32 | 57.54 | 38.97 | 79.06 | 59.70 | 500K |
| FS-M, low rank | **67.88** | **56.05** | **62.65** | 33.16 | **83.10** | **81.10** | 42M |

Table 18: Merging groups of 4 ViT-B-16 with 100 original images from the training set and 800 augmented images from each dataset (resulting in a total of 400 original images and 3,200 augmented images). We report the per-task and joint accuracy on the test set. We will denote: C = Cars, C10 = CIFAR10, C100 = CIFAR100, D = DTD, E = EuroSAT, G = GTSRB, M = MNIST, R = RESISC45, S = SVHN.

| Merging Methods | G, M, C100, S | | E, G, C10, S | | R, E, C10, M | |
|---|---|---|---|---|---|---|
| | Per-task | Joint | Per-task | Joint | Per-task | Joint |
| Original models | 95.09 | - | 97.87 | - | 97.37 | - |
| Ensemble | 85.98 | 40.31 | 91.97 | 58.64 | 89.41 | 50.56 |
| Average | 5.12 | 0.05 | 8.57 | 1.44 | 8.33 | 0.68 |
| SLERP | 6.20 | 0.33 | 7.81 | 1.12 | 8.69 | 0.43 |
| RegMean | 6.32 | 0.25 | 11.29 | 1.54 | 8.33 | 1.23 |
| Opt | 5.17 | 0.16 | 11.53 | 3.90 | 8.76 | 1.62 |
| Distillation | **76.18** | **42.67** | 44.65 | 40.71 | 86.86 | 75.90 |
| FS-Merge, diagonal | 68.87 | 34.59 | 79.30 | 67.62 | 87.35 | 72.17 |
| FS-Merge, low rank | 73.26 | 41.33 | **86.44** | **78.78** | **91.38** | 79.12 |
| FS-Merge seq. | 74.10 | 40.76 | 84.85 | 75.86 | 90.93 | **80.29** |

Table 19: Merging groups of 4 ViT-B-16 with 100 original images from the training set and 800 augmented images from each dataset (resulting in a total of 400 original images and 3,200 augmented images). We report the per-task and joint accuracy on the test set. We will denote: C = Cars, C100 = CIFAR100, D = DTD, E = EuroSAT, G = GTSRB, M = MNIST, R = RESISC45, S = SVHN.

| Merging Methods | C, C100, R, E | | D, M, S, C100 | | E, D, R, M | | #Parameters Optimized |
|---|---|---|---|---|---|---|---|
| | Per-task | Joint | Per-task | Joint | Per-task | Joint | |
| Original models | 91.06 | - | 86.52 | - | 89.05 | - | - |
| Ensemble | 77.56 | 64.78 | 73.72 | 29.98 | 78.58 | 42.25 | - |
| Average | 5.19 | 0.26 | 5.24 | 0.14 | 7.11 | 0.68 | 0 |
| SLERP | 3.15 | 0.19 | 5.37 | 0.25 | 7.47 | 0.59 | 0 |
| RegMean | 4.03 | 0.27 | 5.94 | 0.25 | 7.11 | 0.41 | 0 |
| Opt | 4.17 | 0.23 | 5.92 | 0.26 | 7.94 | 1.67 | 0 |
| Distillation | 59.28 | 53.04 | 66.22 | 36.50 | 53.52 | 45.85 | 111M |
| FS-Merge, diagonal | 60.35 | 46.83 | 60.69 | 29.01 | 56.51 | 46.76 | 600K |
| FS-Merge, low rank | **71.84** | **65.43** | **68.12** | 38.11 | **68.31** | **60.84** | 60M |
| FS-Merge seq. | 71.71 | 61.21 | 67.75 | **38.14** | 66.19 | 56.64 | 18M |

Table 20: Merging groups of 5 ViT-B-16 with 100 original images from the training set and 800 augmented images from each dataset (resulting in a total of 500 original images and 4,000 augmented images). We report the per-task and joint accuracy on the test set. We will denote: C = Cars, C10 = CIFAR10, C100 = CIFAR100, D = DTD, E = EuroSAT, G = GTSRB, M = MNIST, R = RESISC45, S = SVHN.

| Merging Methods | R, M, D, S, C10 | | C, M, C100, E, R | | E, S, C, C10, G | |
|---|---|---|---|---|---|---|
| | Per-task | Joint | Per-task | Joint | Per-task | Joint |
| Original models | 90.25 | - | 92.78 | - | 95.52 | - |
| Ensemble | 75.06 | 35.98 | 76.77 | 50.06 | 82.69 | 65.32 |
| Average | 7.22 | 1.65 | 4.63 | 0.37 | 6.57 | 0.10 |
| SLERP | 7.40 | 0.40 | 6.33 | 0.62 | 6.34 | 0.59 |
| RegMean | 7.08 | 1.91 | 4.64 | 0.67 | 9.16 | 1.57 |
| Opt | 7.67 | 0.65 | 3.98 | 1.69 | 9.18 | 3.18 |
| Distillation | 74.87 | **63.99** | 65.72 | 56.22 | 77.04 | 64.27 |
| FS-Merge, diagonal | 73.30 | 59.82 | 67.16 | 50.49 | 70.27 | 59.00 |
| FS-Merge, low rank | **77.05** | 60.77 | **75.97** | **65.95** | **80.83** | **65.50** |
| FS-Merge seq. | 76.82 | 55.30 | 72.80 | 62.51 | 78.24 | 60.36 |

Table 21: Merging pairs of ViT-L-14 with 100 original images from the training set and 1000 augmented images from each dataset. We report the per-task and joint accuracy on the test set.

| Merging Methods | DTD, EuroSAT | | CIFAR100, SVHN | | Cars, MNIST | |
|---|---|---|---|---|---|---|
| | Per-task | Joint | Per-task | Joint | Per-task | Joint |
| Original models | 81.33 | - | 94.68 | - | 96.50 | - |
| Ensemble | 77.71 | 71.54 | 94.25 | 77.90 | 96.23 | 96.22 |
| Average | 6.36 | 1.40 | 9.18 | 8.03 | 5.98 | 0.08 |
| SLERP | 5.21 | 1.58 | 5.20 | 2.59 | 8.48 | 5.44 |
| RegMean | 8.66 | 4.21 | 5.64 | 0.47 | 10.97 | 0.13 |
| Opt | 10.33 | 3.11 | 4.68 | 2.45 | 5.95 | 5.67 |
| Distillation | 78.51 | **75.84** | 90.91 | 85.78 | 91.82 | 90.58 |
| FS-M | **78.60** | 74.68 | **91.78** | **87.67** | **95.80** | **94.31** |

Table 22: Merging pairs of ViT-L-14 with 100 original images from the training set and 1000 augmented images from each dataset. We report the per-task and joint accuracy on the test set.

| Merging Methods | RESISC45, CIFAR10 | | GTSRB, RESISC45 | | Cars, EuroSAT | |
|---|---|---|---|---|---|---|
| | Per-task | Joint | Per-task | Joint | Per-task | Joint |
| Original models | 96.42 | - | 95.91 | - | 95.71 | - |
| Ensemble | 95.52 | 94.12 | 94.05 | 93.35 | 93.07 | 92.95 |
| Average | 11.21 | 3.83 | 1.92 | 1.10 | 9.36 | 1.97 |
| SLERP | 16.14 | 4.48 | 2.42 | 1.43 | 8.23 | 5.71 |
| RegMean | 13.14 | 8.61 | 4.56 | 2.69 | 6.91 | 6.04 |
| Opt | 10.89 | 3.06 | 3.60 | 2.46 | 3.07 | 0.28 |
| Distillation | 84.37 | 82.20 | **82.62** | **80.64** | 89.11 | 86.98 |
| FS-M diagonal | 82.93 | 77.91 | 76.94 | 73.83 | 91.14 | 82.54 |
| FS-M low rank | **88.02** | **85.46** | 80.54 | 77.13 | **93.85** | **90.61** |

Table 23: Merging pairs of ViT-L-14 with 100 original images from the training set and 1000 augmented images from each dataset. We report the per-task and joint accuracy on the test set.

| Merging Methods | RESISC45, SVHN | | DTD, GTSRB | | CIFAR100, EuroSAT | |
|---|---|---|---|---|---|---|
| | Per-task | Joint | Per-task | Joint | Per-task | Joint |
| Original models | 95.29 | - | 81.22 | - | 95.42 | - |
| Ensemble | 93.53 | 90.37 | 78.22 | 77.02 | 91.13 | 90.46 |
| Average | 8.32 | 1.07 | 2.66 | 1.09 | 9.87 | 0.82 |
| SLERP | 5.97 | 1.65 | 1.80 | 0.72 | 10.48 | 6.29 |
| RegMean | 12.95 | 3.32 | 4.83 | 2.98 | 13.14 | 8.61 |
| Opt | 5.14 | 4.26 | 4.03 | 2.95 | 3.54 | 0.52 |
| Distillation | 87.66 | 77.28 | 67.20 | 65.84 | 92.39 | 86.62 |
| FS-Merge diagonal | 85.93 | 77.27 | 63.79 | 61.36 | 91.99 | 84.80 |
| FS-Merge low rank | **88.11** | **78.32** | **69.09** | **67.14** | **93.56** | **88.20** |

Table 24: Merging pairs of ViT-B-16 models, where the first model is pre-trained using CLIP and the second model is pre-trained on ImageNet. The merge is performed using 16 original images from the training set and 800 augmented images from each dataset. We report the per-task and joint accuracy on the test set.

| Merging Methods | Cars, RESISC45 | | GTSRB, EuroSAT | | CIFAR10, DTD | |
|---|---|---|---|---|---|---|
| | Per-task | Joint | Per-task | Joint | Per-task | Joint |
| Original models | 90.08 | - | 98.99 | - | 81.26 | - |
| Ensemble | 82.40 | 81.94 | 97.72 | 95.18 | 80.58 | 79.88 |
| Average | 1.76 | 1.49 | 6.39 | 2.74 | 6.26 | 4.95 |
| SLERP | 1.50 | 1.14 | 6.11 | 2.60 | 5.89 | 2.26 |
| Opt | 1.87 | 1.20 | 5.75 | 2.18 | 6.10 | 3.61 |
| Distillation | 51.10 | 46.62 | 63.96 | 61.26 | 56.36 | 55.48 |
| FS-Merge | **60.36** | **59.52** | **84.52** | **84.23** | **58.56** | **58.11** |

Table 25: Merging pairs of ViT-B-16 models, where the first model is pre-trained using ImageNet and the second model is pre-trained on CLIP. The merge is performed using 16 original images from the training set and 800 augmented images from each dataset. We report the per-task and joint accuracy on the test set.

| Merging Methods | RESISC45, GTSRB | | DTD, MNIST | | SVHN, CIFAR100 | |
|---|---|---|---|---|---|---|
| | Per-task | Joint | Per-task | Joint | Per-task | Joint |
| Original models | 96.19 | - | 81.86 | - | 93.83 | - |
| Ensemble | 92.96 | 91.14 | 83.01 | 69.70 | 82.38 | 72.74 |
| Average | 2.79 | 1.73 | 3.10 | 1.30 | 8.19 | 0.83 |
| SLERP | 2.45 | 1.43 | 6.59 | 1.43 | 8.08 | 0.48 |
| Opt | 1.86 | 1.02 | 4.75 | 1.32 | 7.21 | 0.47 |
| Distillation | 60.08 | 57.63 | 67.62 | 67.35 | 50.25 | **43.12** |
| FS-Merge | **64.55** | **62.43** | **73.90** | **70.97** | **51.74** | 40.06 |

(Netzer et al., 2011), and EuroSAT (Helber et al., 2019). Additionally, the vision encoder from a pre-trained CLIP model (Radford et al., 2021), which is also a ViT-B-16, was used. This ViT was pre-trained from scratch using a contrastive learning approach that leverages pairs of similar and dissimilar image-caption pairs. Subsequently, this ViT was fine-tuned on Cars (Krause et al., 2013), GTSRB (Stallkamp et al., 2011), MNIST (LeCun, 1998), CIFAR10, and CIFAR100 (Krizhevsky et al., 2009).

Table 24 presents experiments in which a CLIP pre-trained model was merged with an ImageNet pre-trained model, using the CLIP model as the "first" initialization. Table 25 shows similar experiments, using the ImageNet pre-trained model as the "first" initialization. Note that the order of the models is important due to the "first" initialization used by both FS-Merge and distillation. For merging, 16 original images from the training set and 800 augmented images were used from each dataset (a total of 1,632 images per merge). The merged model was then evaluated on the fine-tuned datasets. FS-Merge was used with low rank of 12.

As shown, even with the new pre-training strategy, our main conclusions from the other experiments remain consistent: local and simple merging methods, such as Average and Opt, fail in this setting, while FS-Merge continues to outperform distillation by a significant margin in most cases.

### F.4 Merging Vision Transformers fine-tuned from the same initialization

To explore FS-Merge's capabilities, we investigate its performance in the more common setting found in merging method literature: merging models fine-tuned on different tasks from the same pre-trained initialization. We fine-tuned the ViT-B-16 model on GTSRB (Stallkamp et al., 2011), MNIST (LeCun, 1998),

Table 26: Merging ViT-B-16 models, fine-tuned on different tasks **from the same pre-trained initialization**. The merge is performed using 100 original images from the training set and 1000 augmented images from each dataset. The per-task and joint accuracy on the test set are reported. We will denote: C10 = CIFAR10, C100 = CIFAR100, G = GTSRB, M = MNIST, R = RESISC45.

| Merging Methods | G, R | | R, C10, G | | C100, R, M, G | |
|---|---|---|---|---|---|---|
| | Per-task | Joint | Per-task | Joint | Per-task | Joint |
| Original models | 96.135 | - | 96.68 | - | 94.45 | - |
| Ensemble | 92.66 | 86.85 | 86.15 | 67.88 | 80.40 | 41.91 |
| Average | 82.24 | 72.43 | 59.00 | 37.71 | 41.42 | 12.20 |
| SLERP | 82.23 | 72.42 | 57.08 | 45.68 | 38.76 | 30.82 |
| RegMean | **93.56** | 90.35 | 87.51 | 76.29 | 75.72 | 46.73 |
| Distillation | 91.76 | 90.37 | 90.85 | 86.44 | 82.41 | **63.11** |
| FS-Merge | 93.30 | **91.41** | **91.62** | **86.52** | **84.51** | 58.43 |

Table 27: Merging ViT-B-16 models, fine-tuned on SUN397, Food101 and CIFAR100. The merge is performed using 100 original images from the training set and 1000 augmented images from each dataset. The test accuracies for individual tasks, along with the per-task and joint accuracies on the test set, are reported.

| Merging Methods | SUN397 | Food101 | CIFAR100 | Per-task accuracy | Joint accuracy |
|---|---|---|---|---|---|
| Original models | 65.37 | 81.93 | 85.61 | 77.63 | - |
| Ensemble | 46.91 | 69.13 | 74.96 | 63.66 | 55.61 |
| Average | 0.10 | 0.95 | 0.50 | 0.51 | 0.05 |
| SLERP | 0.15 | 0.95 | 0.74 | 0.61 | 0.32 |
| Opt | 0.12 | 0.90 | 0.65 | 0.55 | 0.24 |
| Distillation | 44.20 | 20.46 | 35.19 | 33.28 | 27.17 |
| FS-Merge | **47.71** | **26.43** | **38.87** | **37.67** | **32.82** |

RESISC45 (Cheng et al., 2017), CIFAR10, and CIFAR100 (Krizhevsky et al., 2009), all starting from the same pre-trained model. Subsequently, we merged groups of these fine-tuned models in varying sizes.

The results, shown in Table 26, utilize FS-Merge and distillation with the "average" initialization, as this configuration yielded better accuracy on the validation set. As demonstrated, FS-Merge outperforms other baselines in most cases. However, we note that in this setting, FS-Merge is less dominant, as its margin over other baselines, such as distillation and RegMean, is relatively small. Moreover, when the models are fine-tuned from the same pre-trained model, traditional merging methods, such as RegMean, achieve performance very close to FS-Merge while using significantly fewer resources. Thus, despite its success in this scenario, we conclude that FS-Merge's true strength lies in more challenging scenarios involving models with different initializations.

### F.5 Merging Vision Transformers on large datasets

To evaluate the merging methods on larger datasets, different pre-trained models were fine-tuned on SUN397 Xiao et al. (2010), Food101 (Bossard et al., 2014), and CIFAR100 (Krizhevsky et al., 2009). Combined, these three datasets contain over 290,000 images and 598 classes. This represents the most challenging merging attempt known to us in the literature under our setting (merging differently initialized transformers with only a small unlabeled subset of the training data). Table 27 presents the results of merging these models using 100 original images and 100 augmented images per dataset. FS-Merge was applied with a low rank of 32. As shown, FS-Merge outperforms the baselines even in this challenging scenario.

### F.6 Merging large number of Vision Transformers

To investigate the ability of FS-Merge Seq. in the extremely challenging setting of merging a large number of models, we merged all our existing ViT-B-16 models, which were trained from different initializations on different tasks—a total of eight models. We used a relatively small amount of data: 100 original images and 1000 augmented images per task. The models were trained on Cars, DTD, CIFAR100, GTSRB, RESISC45, MNIST, EuroSAT, and SVHN, which together comprise around 5000 classes.

As shown in Table 28, this task is difficult, even for the ensemble method, which has direct access to all eight models. FS-Merge Seq. outperforms the baselines and even achieves higher joint accuracy than the ensemble, demonstrating its superiority.

However, we note that a key challenge in this scenario is the computational cost of merging such a large number of models: FS-Merge was not evaluated due to its high memory requirements for storing the frozen weights of all models. Additionally, the KD process took around 4.0 hours, while FS-Merge Seq. required approximately 5.3 hours. In contrast, traditional merging methods execute much faster: averaging and SLERP take only a few seconds, RegMean takes around 6 minutes, and Opt requires approximately 41 minutes. However, all of these methods are inadequate in this challenging setting, resulting in merged models that perform no better than random guessing.

Table 28: Merging eight ViT-B-16 models trained on different tasks using 100 original and 1000 augmented images per task. We report per-task and joint accuracy on the test set.

| Metric | Ensemble | Average | SLERP | Opt | RegMean | Distillation | FS-Merge seq. |
|---|---|---|---|---|---|---|---|
| Per-task accuracy | 70.53 | 4.44 | 5.45 | 4.78 | 5.03 | 63.72 | **68.81** |
| Joint accuracy | 42.37 | 0.23 | 0.17 | 0.11 | 0.17 | 42.98 | **46.00** |

### F.7 Merging text Transformers

In this series of experiments, we aimed to evaluate FS-Merge on a different modality: merging differently initialized text transformers. Following (Verma & Elbayad, 2024), we used five bert-base-uncased models from the MultiBERTs reproductions (Devlin et al., 2018; Sellam et al., 2022), each trained from a different random initialization and with a different data ordering. We fine-tuned each model on a distinct classification task from the GLUE dataset (Wang et al., 2019), using six tasks: RTE, QQP, MNLI, MRPC, SST-2, and QNLI. The same pre-trained BERT was fine-tuned on QQP and MRPC, so their merge was not evaluated. It is worth noting that ViT is a pre-LN Transformer, while BERT is a post-LN Transformer (Xiong et al., 2020). Moreover, unlike the ViT experiments where models were fine-tuned with frozen classification heads derived from CLIP embeddings, these experiments used trainable classification heads.

**Baselines.** We compared our method against two baselines: "average" (Wortsman et al., 2022), a simple weight averaging technique; and distillation (Hinton et al., 2015), which trains a single BERT to mimic the features of the last representation layer of the original models. We also reported the performance of the "original models", representing the average accuracy of the models to be merged; and ensemble (Ganaie et al., 2022), which averages the models' outputs and then applies classification heads. Note that these last two are not valid merging methods as they use the original models directly.

**Hyperparameters.** We performed a hyperparameter search similar to the ViT case (Appendix E.2). We selected a pair of tasks, QQP and MNLI, and created a validation set for each task from the training set. We then fine-tuned a pair of differently initialized BERT models, one for each task. These two models were used to conduct a hyperparameter search for both distillation and FS-Merge using the same hyperparameter grid. The hyperparameters that maximized the average per-task validation accuracy for this pair were chosen and then applied when merging the other BERT pairs. For FS-Merge, the chosen hyperparameters were 400 epochs, learning rate of 0.0001, weight decay of 0, "first" initialization, and low rank of 12. For Distillation, the chosen hyperparameters were 300 epochs, learning rate of 0.0001, weight decay of 0, and "first" initialization. Both methods used batch size of 128.

200 data points were taken from each training set to create features for the merging methods. In Table 29, we report the per-task test accuracy of these experiments. We can see that, similarly to the ViT case, traditional merging methods like "average" fail on the challenging task of merging text transformers from different initializations. FS-Merge achieve SOTA results, outperforming the ensemble in some cases.

Table 29: Merging pairs of BERTs with 200 original texts from the training set of each dataset. FS-Merge was used with a low rank of 12. We report the per-task accuracy on the test set.

| Tasks | Original models | Ensemble | Average | Distillation | FS-Merge |
|---|---|---|---|---|---|
| RTE, QQP | 79.49 | 71.13 | 41.87 | 65.00 | **68.88** |
| MNLI, MRPC | 85.73 | 67.17 | 50.52 | 76.87 | **78.29** |
| MRPC, QNLI | 89.02 | 87.19 | 40.53 | **80.67** | 79.33 |
| SST-2, RTE | 80.15 | 71.64 | 51.06 | 75.45 | **76.00** |
| MNLI, SST-2 | 88.39 | 84.78 | 43.75 | 82.81 | **83.24** |
| RTE, QNLI | 79.75 | 74.60 | 48.94 | 66.40 | **69.69** |
| QNLI, QQP | 91.01 | 71.63 | 56.32 | 82.40 | **84.75** |
| SST-2, QNLI | 91.68 | 90.44 | 49.38 | 82.92 | **84.39** |
| MRPC, RTE | 77.49 | 68.41 | 57.83 | **68.70** | 68.37 |
| QNLI, MNLI | 87.99 | 71.83 | 41.62 | 71.83 | **73.79** |
| QQP, RTE | 79.49 | 71.13 | 41.87 | 73.30 | **75.62** |
| MRPC, MNLI | 85.73 | 67.17 | 50.52 | **62.90** | 62.81 |
| QQP, SST-2 | 91.41 | 86.59 | 55.48 | 87.49 | **87.94** |
| MNLI, QQP | 87.7 | 78.69 | 36.09 | 78.74 | **79.22** |
| QNLI, MRPC | 89.02 | 87.19 | 40.53 | 82.97 | **82.98** |
| RTE, SST-2 | 80.15 | 71.64 | 51.06 | 69.20 | **73.39** |
| QNLI, SST-2 | 91.68 | 90.44 | 49.38 | 87.27 | **88.09** |
| RTE, MRPC | 77.49 | 68.41 | 57.83 | 63.46 | **66.05** |
| SST-2, MNLI | 88.39 | 84.78 | 43.75 | 66.44 | **67.29** |
| QQP, QNLI | 91.01 | 71.63 | 56.32 | 82.16 | **82.79** |

### F.8  Number of original training images - Vision Transformer

Here we present additional experiments which examine the effect of the number of original images versus augmented images on the ViT merged model. Original images, defined as those sourced from the training datasets of the models to be merged, were varied in number: 16, 64, 128, 256, 512, 1024. For each scenario, augmented images were generated to bring the total images per dataset to 1024, ensuring uniform dataset size across all cases. These images were then used to create features for the merging processes.

Ensemble, Distillation, diagonal FS-Merge (low rank of 0), and FS-Merge with low rank of 24 were applied to merge pairs of ViT-B-16 models. It should be noted that Ensemble is not considered a legitimate merging method. The per-task and joint accuracy on the test set were reported, employing the optimal hyperparameters identified in earlier chapters. The outcomes of this experiment are presented in Figure 5 and Figure 6. Note that in these experiments, different features were used compared to the old experiments, so the results may vary.

### F.9  Number of training data points - BERT

We examined the impact of the amount of data points taken on the merged BERT model. Training data points, sourced from the training datasets of the models to be merged, were adjusted to sizes of: 16, 64, 128, 256, 512, 1024. These samples were used to generate features for the merging process. Unlike the ViT case, we did not employ data augmentation, resulting in non-uniform dataset sizes in each case. To compensate, an additional 200 data points from each dataset were used to implement early stopping, preventing overtraining with smaller datasets.

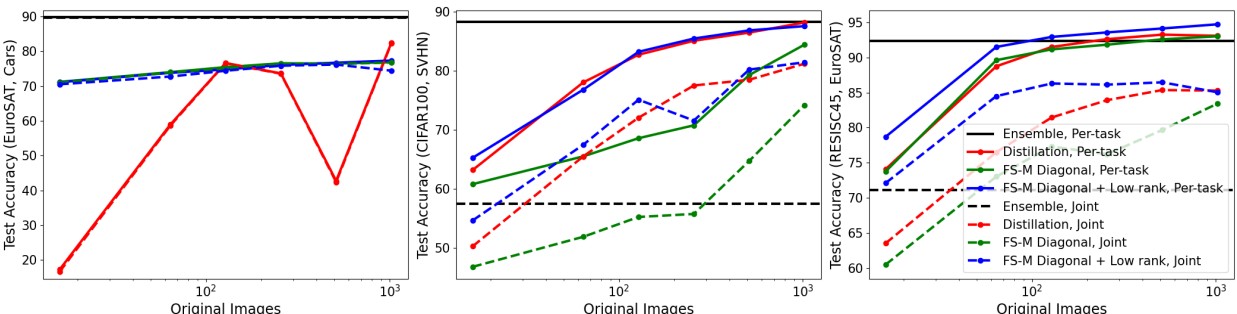

Figure 5: We used Ensemble, Distillation, and FS-Merge to merge pairs of models trained on EuroSAT and Cars (left), CIFAR100 and SVHN (center), RESISC45 and EuroSAT (right). We varied the number of original images per dataset and added augmentation images so the total number of images per dataset would be 1024. We present the per-task and joint accuracy.

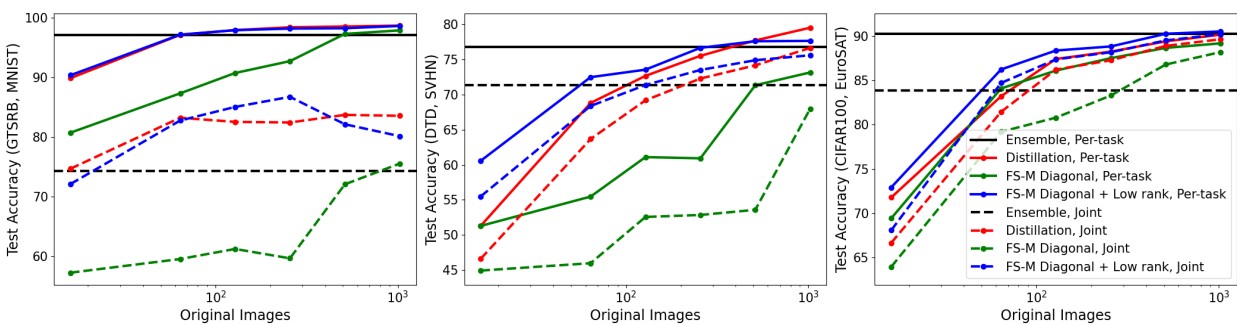

Figure 6: We used Ensemble, Distillation, and FS-Merge to merge pairs of models trained on GTSRB and MNIST (left), DTD and SVHN (center), CIFAR100 and EuroSAT (right). We varied the number of original images per dataset and added augmentation images so the total number of images per dataset would be 1024. We present the per-task and joint accuracy.

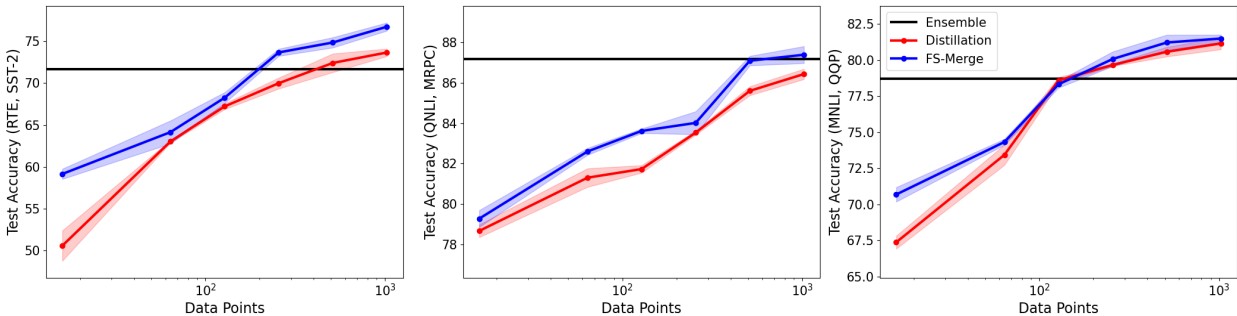

Figure 7: Merging of BERT model pairs trained on RTE and SST-2 (left), QNLI and MRPC (center), MNLI and QQP (right) using Ensemble, Distillation, and FS-Merge. The graph shows per-task test accuracy across varying training data sizes. Three runs were conducted, each with a different random seed, to generate error bars and verify the statistical significance of the results.

We applied Ensemble, Distillation, and FS-Merge with a low rank of 12 to merge pairs of BERT base models. It is important to note that Ensemble is not considered a valid merging method. We reported per-task accuracy on the test set using optimal hyperparameters identified previously. The results of this experiment are shown in Figures 7.

## G   Ablation studies

### G.1   FS-Merge ablation study

Table 30 shows an ablation study of merging a group of three ViT-B-16 using FS-Merge. This study involved three groups of models. We evaluated both the per-task average accuracy and joint accuracy on the test set.

We first used FS-Merge diagonal, where the $M, U$ were parameterized as a concatenation of diagonal matrices. We used random initialization, and created features using only the original 100 images from each dataset without augmentations (so a total number of 300 images). Then, 1000 more augmented images were added for each dataset. Then, "average initialization" for the $M, U$ was tested, meaning the Foldable SuperNet initialized the merged model from the average of the original ones; and also first initialization was tested, so the Foldable SuperNet initialized the merged model from the first original model. "Low rank" stand for FS-Merge with $M, U$ parametrized as a concatenation of diagonal matrices plus low-rank matrices (Eq. 7), with low rank of 24.

As can be observed, the "first" initialization leads to a better merged model compared to the average or random initialization. Additionally, this experiment shows the significance of creating more images through augmentations and using low rank in the $M, U$ matrices of the Foldable SuperNet.

Table 31 displays the per-task accuracy for each task of the merged model across RESISC45, CIFAR10, and EuroSAT. It can be seen that the "first" initialization improves not only the first task's accuracy but also the accuracy across all tasks. This behavior recurs in every other group of models that we merged.

Our experiments suggest that in the ViT case, initialization is extremely important for training the Foldable SuperNet. When initialized randomly, FS-Merge does not converge, even with the addition of augmented data (and see Appendix G.4 discussing this issue). Only Foldable SuperNet initialized from the average or first model allows it to converge into a successful merged model.

### G.2   Knowledge Distillation ablation study

Table 32 presents an ablation study on merging a group of three ViT-B-16 models using distillation. This study involved the same three groups of models used in the FS-Merge ablation study (Table 30). We followed

Table 30: Ablation study comparing the effectiveness of different initialization and augmentation strategies on the merging of groups of three ViT-B-16. Only our method, FS-Merge, was used. We show the per-task and joint accuracy on the test set. "Aug" stands for Augmentations, and "Init" stands for Initialization. We will denote: C = Cars, C10 = CIFAR10, C100 = CIFAR100, D = DTD, E = EuroSAT, G = GTSRB, M = MNIST, R = RESISC45, S = SVHN.

| Setting | | | R, C10, E | | S, E, G | | C10, G, M | |
|---|---|---|---|---|---|---|---|---|
| FS-Merge | Init | Aug | Per-task | Joint | Per-task | Joint | Per-task | Joint |
| Diagonal | Random | ✗ | 7.43 | 0.77 | 11.05 | 1.18 | 7.71 | 1.18 |
| Diagonal | Average | ✗ | 25.87 | 10.15 | 21.42 | 12.19 | 14.71 | 4.25 |
| Diagonal | First | ✗ | 76.75 | 55.44 | 66.87 | 50.20 | 67.74 | 51.38 |
| Low rank | First | ✗ | 83.40 | 73.76 | 77.48 | 65.69 | 80.77 | 64.93 |
| Diagonal | First | ✓ | 86.04 | 73.47 | 82.15 | 71.18 | 76.45 | 57.13 |
| Low rank | First | ✓ | **89.17** | **79.34** | **85.96** | **75.38** | **86.41** | **66.84** |

Table 31: Merging three ViT-B-16 models, fine-tuned on RESISC45, CIFAR10 and EuroSAT, using diagonal FS-Merge. We are examining the effects of initialization and augmentation on the per-task accuracy of the test set.

| FS-Merge details | | | Original tasks | | |
|---|---|---|---|---|---|
| Initialization | Augmentations | Average Acc | RESISC45 | CIFAR10 | EuroSAT |
| Average | ✗ | 25.87 | 3.73 | 22.94 | 50.94 |
| First | ✗ | 76.75 | **89.4** | 56.24 | 84.62 |
| Average | ✓ | 37.51 | 7.3 | 33.59 | 71.64 |
| First | ✓ | **86.04** | 87.86 | **79.76** | **90.52** |

Appendix G.1, and created features using 100 original images and 1000 augmented images from each dataset (resulting in a total of 3,300 images).

We aimed to investigate how different initializations affect the performance of the distillation merge, also examining traditional merging baselines as initializations (such as average and RegMean). Additionally, we explored the impact of augmentations. We evaluated both the per-task average accuracy and the joint accuracy on the test set.

Table 32: Ablation study comparing the effectiveness of different initialization and augmentation strategies on the merging of groups of three ViT-B-16. Only distillation merge was used. We show the per-task and joint accuracy on the test set. We will denote: C = Cars, C10 = CIFAR10, C100 = CIFAR100, D = DTD, E = EuroSAT, G = GTSRB, M = MNIST, R = RESISC45, S = SVHN.

| Distillation | | R, C10, E | | S, E, G | | C10, G, M | |
|---|---|---|---|---|---|---|---|
| Initialization | Augmentations | Per-task | Joint | Per-task | Joint | Per-task | Joint |
| Random | ✗ | 33.20 | 25.93 | 29.00 | 25.75 | 27.56 | 19.77 |
| Random | ✓ | 39.64 | 30.83 | 34.39 | 30.13 | 33.17 | 24.69 |
| Average | ✗ | 38.16 | 30.22 | 37.78 | 33.49 | 37.40 | 25.68 |
| Average | ✓ | 41.78 | 32.76 | 55.35 | 46.20 | 52.84 | 39.54 |
| RegMean | ✗ | 49.44 | 39.61 | 66.50 | 55.54 | 57.80 | 46.31 |
| RegMean | ✓ | 58.53 | 47.84 | 77.18 | 66.42 | 72.55 | 57.42 |
| First | ✗ | 81.56 | 71.32 | 77.24 | 65.02 | 78.38 | 67.76 |
| First | ✓ | **84.14** | **73.98** | **84.46** | **73.19** | **84.30** | **70.92** |

Table 33: Comparing Distillation and FS-Merge (both with "first" init), with and without augmentations, while merging groups of three ViT-B-16. Features were created using 100 original images and 1,000 augmented images per dataset. We show the per-task and joint accuracy on the test set. We will denote: C = Cars, C10 = CIFAR10, C100 = CIFAR100, D = DTD, E = EuroSAT, G = GTSRB, M = MNIST, R = RESISC45, S = SVHN.

| Setting | | R, C10, E | | S, E, G | | C10, G, M | |
|---|---|---|---|---|---|---|---|
| Method | Aug | Per-task | Joint | Per-task | Joint | Per-task | Joint |
| Distillation | $\times$ | 81.56 | 71.32 | 77.24 | 65.02 | 78.38 | **67.76** |
| FS-Merge, Low rank | $\times$ | **83.40** | **73.76** | **77.48** | **65.69** | **80.77** | 64.93 |
| Distillation | $\sqrt{}$ | 84.14 | 73.98 | 84.46 | 73.19 | 84.30 | **70.92** |
| FS-Merge, Low rank | $\sqrt{}$ | **89.17** | **79.34** | **85.96** | **75.38** | **86.41** | 66.84 |

As observed, the "first" initialization leads to a superior merged model compared to all other initializations, including other merging baselines such as average and RegMean. Moreover, augmentations enhance performance in all cases.

In Table 33, we compare the best distillation and FS-Merge versions from the ablation studies (i.e., with "first" initialization), showing that FS-Merge outperforms distillation with and without augmentations.

### G.3    Merging Vision Transformers: Local VS. Global perspective

Observe that in the MLP merging method from Section 2.1, the local version of FS-Merge was used, which involves training a Foldable SuperNet individually for each layer. In the ViT case, local FS-Merge involves training each block of the Foldable SuperNet separately. For example, training a Foldable SuperNet that merges the first attention blocks, then training a Foldable SuperNet that merges the first MLP blocks, and so on. We found empirically that this approach leads to a poor solution in the ViT case, resulting in a dysfunctional merged model with accuracy nearly as poor as a random guess. Instead, we found that the global version of FS-Merge is much more effective in this case, involving training the entire Foldable SuperNet of the ViT to reconstruct the features of the last representation layer of the original models, $f_L$.

A few explanations exist for this issue. First, training the Foldable SuperNet in a local manner for the ViT, meaning block-wise, must be performed on very unnatural blocks, which "break" the transformer blocks. This is necessary because $M$ and $U$ matrices must be placed before or after a linear layer to allow them to be folded after training. Moreover, the attention score computation, layer normalization, and skip connections must be performed on the merged features (with the lower dimensionality). These conditions forced the "breaking" of existing ViT blocks, and, for example, required teaching the Foldable SuperNet of the attention block to reconstruct features that are within the next MLP blocks. This complicated structure probably have hindered the optimization process.

Second, the Foldable SuperNet consistently uses the merged features in the skip connection. This means that when learning block-wise, the features forwarded via skip connection to the next block are dramatically changed. These new merged features are very different from the original ones, which likely severely affected the optimization of the next Foldable SuperNet block.

### G.4    Merging Vision Transformer with randomly initialized Foldable SuperNet

In our experiments, we found that smart initialization of the Foldable SuperNet is crucial for FS-Merge. As common in NNs, we first tried to initialize the $M, U$ matrices using a random Gaussian distribution dependent on the hidden dimension (He et al., 2015).

In the MLP case, a random initialization can work, but better results are achieved when using ZipIt as the initialization method for the Foldable SuperNet (and see Section 3.1). In the ViT case, We tested multiple

scales for the random initialization, but could not find a setup that allowed the FS-Merge to converge into a functional merged model.

This led us to study smarter initializations, such as "average". When merging $K$ models, a Foldable SuperNet that creates an average merge of the weights is achieved by setting all the $M, U$ matrices as follows:

$$M = \begin{pmatrix} \frac{I}{K} \\ ... \\ \frac{I}{K} \end{pmatrix}, \ U = \begin{pmatrix} I & ... & I \end{pmatrix} .$$

When $I$ is the identity matrix. In the case of ViTs, the "first" initialization proved to be the most effective, involving initializing the Foldable SuperNet so it exclusively selects the weights of the first model. It can be achieved by setting all the $M, U$ matrices as follows:

$$M = \begin{pmatrix} I \\ 0 \\ ... \\ 0 \end{pmatrix}, \ U = \begin{pmatrix} I & ... & I \end{pmatrix} .$$

By the end of the training, the "first" initialization results in a merged model with improved accuracy across all tasks, not just the task of the first model (see Appendix G.1 for more details). Surprisingly, averaging initialization also performed well, despite the fact that the average merge is not an effective merging method when combining ViTs trained from different initializations.

This effectiveness is the reason the Foldable SuperNet's $M, U$ matrices were modeled as a sum of low-rank matrices plus a concatenation of diagonal matrices (rather than just a low-rank matrix as in LoRa (Hu et al., 2022)). The concatenation of diagonal matrices enables the initialization of the $M, U$ matrices using those successful methods ("first", "average").

### G.5 Using inner features when merging Vision Transformers

In line with several distillation studies (Wu et al., 2021a; Zagoruyko & Komodakis, 2017; Heo et al., 2019b;a; Park & Kwak, 2019; Liu et al., 2020; Wu et al., 2021b; 2023; Han et al., 2024), we tried to use the inner features of the models to be merged as a regularization for FS-Merge and for distillation. We focused on the features obtained after the MLP block or after the attention block. The attention features in the $l$ block of model $k$, created from the input $I_{\text{img}}^k$, can be written as $f_{l,\text{att}}^k(I_{\text{img}}^k) \in \mathbb{R}^{T \times d}$. Then, the "inner loss" for the global FS-Merge, when handling two tasks $A$ and $B$, can be defined as follows:

$$L = L_{\text{out}} + \lambda \sum_{l \in C} \mathbb{E}_{I_{\text{img}}^A \sim D^A} \left\| f_{l,\text{att}}^A(I_{\text{img}}^A) - \tilde{f}_{l,\text{att}}(I_{\text{img}}^A)[A] \right\|_2^2 +$$

$$\mathbb{E}_{I_{\text{img}}^B \sim D^B} \left\| f_{l,\text{att}}^B(I_{\text{img}}^B) - \tilde{f}_{l,\text{att}}(I_{\text{img}}^B)[B] \right\|_2^2 .$$

Where $L_{\text{out}}$ is the regular global reconstruction loss, which attempts to reconstruct the features of the two original models from the layer preceding the classification head (Appendix B.2). $\tilde{f}_{l,\text{att}}[k]$ are the reconstructed attention features of our Foldable SuperNet in block $l$ for model $k$. $C$ is the set of blocks from which we decided to extract features, and $\lambda$ is the regularization coefficient. $D^A$ and $D^B$ are defined as small subsets of data from the training data of tasks $A$ and task $B$ respectively.

This can easily be rewritten for distillation, only that for each $k$ we compare $f_{l,\text{att}}^k$ with $\tilde{f}_{l,\text{att}}$, which are the inner features of the student model in the matching block. Note that in the distillation method, this regularization forces the inner features $\tilde{f}_{l,\text{att}}$ to resemble the inner features of all the models to be merged (in our example, both $A$ and $B$). In contrast, in FS-Merge, after applying the $U$ matrix, the reconstruction of features from both models $A$ and $B$ are obtained, and each set of these features will be compared with its corresponding ground truth.

Table 34: The effect of inner loss on distillation, FS-Merge low rank, and FS-Merge full rank is demonstrated. Pairs of ViT-B-16, fine-tuned on RESISC45 and CIFAR10, were merged using 100 original images and 1000 augmented images per task. The per-task accuracy on the test set is presented.

| Method | Inner loss details | Per-task Accuracy |
|---|---|---|
| Distillation | $\lambda = 0$ | 86.41 |
| Distillation | $\lambda = 0.1, n = \{5\}$ | 83.22 |
| FS-Merge, rank 12 | $\lambda = 0$ | **89.78** |
| FS-Merge, rank 12 | $\lambda = 0.1, n = \{5\}$ | 86.39 |
| FS-Merge, full rank | No regularization | 40.20 |
| FS-Merge, full rank | $\lambda = 0.1, n = \{5\}$ | 52.77 |
| FS-Merge, full rank | $\lambda = 0.1, n = \{3, 5, 7, 9\}$ | 60.41 |
| FS-Merge, full rank | $\lambda = 0.5, n = \{3, 5, 7, 9\}$ | 70.52 |
| FS-Merge, full rank | $\lambda = 1, n = \{3, 5, 7, 9\}$ | 70.11 |

We tried multiple $\lambda$ values, and various $C$ sets, for features from the MLP or attention blocks. Yet, it seems to only detriment the performance of FS-Merge and distillation. Table 34 shows some of those experiments, merging pairs of ViT-B-16, fine-tuned on RESISC45 and CIFAR10, using 100 original images and 1000 augmented images per task.

It should be observed that specifically in the case of FS-Merge using full rank $M$ and $U$ matrices, the inner loss seems to improve the results. We conclude this because, in the full $M, U$ case, there is a very large number of learnable parameters, so a stronger regularization is needed. However, using full rank $M, U$ matrices in the ViT case is not recommended due to the very high memory and time complexity, and even with this regularization, it underperforms compared to Low rank FS-Merge.

# H  FS-Merge expressive power

As shown in Section 3.2, Section 3.3, and Appendix F.2, FS-Merge significantly outperforms traditional merging methods, such as weight averaging and RegMean, when merging transformers trained from different initializations and tasks. In order to explain this, we present theoretical results that highlight FS-Merge's superior expressive power compared to the baselines.

We examine the case of merging a pair of Multi-Layer Perceptron (MLP) models, $A$ and $B$. Let us consider the $l$-th layer in the MLP model $A$. The features at this layer, denoted by $f_l^A \in \mathbb{R}^{d \times N}$, can be expressed as follows:

$$z_l^A = W_l^A f_{l-1}^A \, , \ f_l^A = \sigma(z_l^A) \, . \tag{8}$$

Here, $W_l^A \in \mathbb{R}^{d \times d}$ are the weights of the current linear layer, $d$ denotes the width, $N$ denotes the number of data points, $\sigma$ represents the ReLU activation function ($\sigma(x) = \max(0, x)$), and $z_l^A \in \mathbb{R}^{d \times N}$ are the pre-activation features. And similarly for model $B$.

**Definition H.1.** A matrix $A \in \mathbb{R}^{d_1 \times d_2}$ is said to have rank $r \leq d_1, d_2$ if the dimension of its column space (or equivalently, its row space) is $r$. It implies that the maximum number of linearly independent columns (or rows) in $A$ is $r$.

**Definition H.2.** Let $x$ be a data point sampled from the distribution $\mathcal{D}$, and let $f_l(x)$ denote the features of the $l$-th layer of an MLP computed from these inputs. We define the features $f_l$ as having a low rank of $r > 0$ if $f_l(x)$ has a rank of at most $r$ for any $x \sim \mathcal{D}$.

We will assume low-rank features in several of the following sections, as it has been found that neural network features are often effectively low-rank Cai et al. (2021); Yu & Wu (2023). Such low-rank features imply that it is possible to use low-rank weights to achieve the same function (i.e., the input-output relation) of the model, even if the original weights are high-rank. In other words, if $f_l$ is a set of features with rank $r$, even if $W_l$ is full rank, we can find a different matrix $\bar{W}_l$ of rank $r$ such that

$$\bar{W}_l f_l = W_l f_l \, . \tag{9}$$

This holds because the action of a linear transformation (such as $W_l$) on a subspace can always be reproduced by another transformation of the same rank as the subspace.

Our main theoretical result, presented in Theorem H.3, demonstrates the significant expressive power of both full-rank and low-rank FS-Merge, surpassing that of the two popular merging methods, Git Re-Basin Ainsworth et al. (2023) and RegMean Jin et al. (2023).

**Theorem H.3.** *A summary of our theoretical results:*

1. *Given two MLP models, $A$ and $B$, where at each layer at least one has an invertible weight matrix, a full-rank FS-Merge can reconstruct any target MLP model of the same size from these models. The same holds for a low-rank FS-Merge, assuming that the target MLP features are low-rank.*

2. *Any merged MLP obtained using the RegMean or Git Re-Basin merging methods can also be constructed using full-rank FS-Merge, even without assuming invertible weights. Moreover, there are scenarios that FS-Merge can solve which RegMean or Git Re-Basin cannot. The same holds for low-rank FS-Merge, given the additional assumption of low-rank features.*

As shown in Theorem H.3, full-rank FS-Merge has greater expressiveness compared to low-rank FS-Merge. However, we argue that in the context of merging large models with a limited amount of data, low-rank FS-Merge is preferable, as it introduces fewer learnable parameters and thus enhances generalization. This claim is supported by our experiments, which show that low-rank FS-Merge outperforms full-rank FS-Merge (Appendix G.5) and Knowledge Distillation (Appendix 3.2) when merging transformers.

*Proof.* Part 1 of Theorem H.3 is proved in Appendix H.1 below.

Part 2 of Theorem H.3 follows from Proposition H.4 and Lemma H.5, which are proven in Appendix H.2 and Appendix H.3, respectively.

**Proposition H.4.** *Given the linear layers of two models, A and B:*

1. *Any merged model obtained using RegMean or Git Re-Basin can also be constructed using full-rank FS-Merge.*

2. *Assuming that the features of at least one model have a low rank of $r$, then for any solution created by Git Re-Basin, low-rank FS-Merge with a rank of $r$ can create an equivalent solution.*

3. *Assuming that the features of both models have a low rank of $r$, any merged model obtained using RegMean can also be constructed using low-rank FS-Merge with a rank of $2r$.*

**Lemma H.5.** *Consider the scenario of two linear layers that share the same backbone weight matrix, with each applying a different feature selection, permutation, and scaling matrix. Then, full-rank and low-rank FS-Merge can merge these layers into a single layer in a manner that allows them to reconstruct the original features from the merged one. In contrast, RegMean and Git Re-Basin cannot perfectly merge these two layers.*

$\square$

## H.1 Proof of Part 1 of Theorem H.3: FS-Merge expressive power

The Foldable SuperNet for two linear layers is defined as follows:

$$\tilde{f}_l(z_l^A, z_l^B) = U_l \sigma(M_l(z_l^A \,||\, z_l^B)), \tag{10}$$

where the input is a concatenation of the original pre-activation features $z_l^A \,||\, z_l^B \in \mathbb{R}^{2 \cdot d \times N}$, and $\tilde{f}_l \in \mathbb{R}^{2 \cdot d \times N}$ are the features reconstruction attempt. Intuitively, FS-Merge learns to compress and reconstruct the features of two different models, while also taking into account the activation function.

In the full rank vesion of FS-Merge, $M_l \in \mathbb{R}^{d \times 2 \cdot d}$ and $U_l \in \mathbb{R}^{2 \cdot d \times d}$ are the learnable parameters of the Foldable SuperNet. In the low rank version of FS-Merge, the $M_l$ and $U_l$ matrices are parametrized as a sum of a low-rank matrix and a concatenation of diagonal matrices. For example, in the case of $M_l$:

$$M_l = M_l^{\mathrm{diag}} + M_l^1 M_l^2. \tag{11}$$

When $r$ is the rank, $M_l^1 \in \mathbb{R}^{d \times r}$, $M_l^2 \in \mathbb{R}^{r \times 2 \cdot d}$, and $M_l^{\mathrm{diag}} \in \mathbb{R}^{d \times 2 \cdot d}$ is a concatenation of two diagonal matrices, each with $d$ learnable parameters. A similar structure is proposed for $U$. In both version of FS-Merge, the merged layer is created as follows:

$$W_l = M_l \begin{pmatrix} W_l^A & 0 \\ 0 & W_l^B \end{pmatrix} U_{l-1}. \tag{12}$$

Next, we wish to prove part 1 of Theorem H.3

*Proof.* Let $l$ be an arbitrary layer in the MLPs. Given two linear weight matrices $W_l^A, W_l^B \in \mathbb{R}^{d \times d}$, we will assume without loss of generality that $W_l^A$ is invertible. Let $W_l^*$ denote the target MLP weights. In the case where the input features for the weight $W_l^*$ are full rank, full-rank FS-Merge is capable of reconstructing the target weights using:

$$M_l = \begin{pmatrix} W_l^*(W_l^A)^{-1} & 0 \end{pmatrix}, U_l = \begin{pmatrix} I \\ I \end{pmatrix}. \tag{13}$$

As the merged weight will be

$$W_l = \begin{pmatrix} W_l^*(W_l^A)^{-1} & 0 \end{pmatrix} \begin{pmatrix} W_l^A & 0 \\ 0 & W_l^B \end{pmatrix} \begin{pmatrix} I \\ I \end{pmatrix} = W_l^*. \tag{14}$$

In the case where the input features for the weight $W_l^*$ have a low rank of $r$, there exists an equivalent weight $\bar{W}_l^*$ with rank $r$ that can generate the same features (i.e. $\bar{W}_l^* f_l = W_l^* f_l$). In this case, low-rank FS-Merge with rank $r$ can reconstruct these equivalent weights using a similar solution:

$$M_l = \begin{pmatrix} \bar{W}_l^*(W_l^A)^{-1} & 0 \end{pmatrix}, U_l = \begin{pmatrix} I \\ I \end{pmatrix}. \tag{15}$$

This solution can be achieved using a low-rank FS-Merge, as $U_l = \begin{pmatrix} I \\ I \end{pmatrix}$ can be reconstructed using its diagonal component $U_l^{\mathrm{diag}}$. Additionally, the matrix $\begin{pmatrix} \bar{W}_l^*(W_l^A)^{-1} & 0 \end{pmatrix}$ has rank $r$ or lower, meaning it can be reconstructed using the low-rank component $M_l^1 M_l^2$.

$\square$

This result happens generically, since we only require that for each layer, one of the models (either $A$ or $B$) has a full-rank weight matrix. This happens almost surely with respect to the Lebesgue measure, as almost all randomly chosen weight matrices will be full rank (i.e., singular matrices have measure zero).

### H.2 Proof of Proposition H.4: FS-Merge expressiveness compared to RegMean and Git Re-Basin

Here we aim to demonstrate that full-rank FS-Merge can generate any solution produced by the popular merging methods Git Re-Basin Ainsworth et al. (2023) and RegMean Jin et al. (2023), even without assuming an invertible weight matrix. The same applies to low-rank FS-Merge, with a few additional assumptions.

Git Re-Basin Ainsworth et al. (2023) is an alignment-based merging method that aligns linear layers using a permutation matrix before averaging them. It defines the activation matching objective for the $l$-th layer as:

$$P_l = \underset{P_l \in S_d}{\operatorname{argmin}} \left\| f_l^A - P_l f_l^B \right\|_2^2, \tag{16}$$

Where $S_d$ is the set of all permutation matrices of size $d \times d$. A permutation matrix is a square binary matrix that has exactly one entry of 1 in each row and each column, with all other entries being 0. It then merges the weights as follows:

$$W_l = \frac{1}{2} W_l^A + \frac{1}{2} P_l W_l^B P_{l-1}^\top. \tag{17}$$

Intuitively, Git Re-Basin searches for a permutation matrix $P_l \in S_d$ that makes the features of the linear layers $f_l^A$ and $f_l^B$ as similar as possible (in terms of the Frobenius norm). Applying this permutation to the linear layer weights $W_l^B$ is equivalent to rearranging the neurons in the layer so that neurons generating similar features are aligned in the same positions.

RegMean Jin et al. (2023) is a strong averaging-based merging method, targeting the following objective:

$$W_l = \underset{W_l}{\operatorname{argmin}} \left\| W_l f_{l-1}^A - W_l^A f_{l-1}^A \right\|_2^2 + \left\| W_l f_{l-1}^B - W_l^B f_{l-1}^B \right\|_2^2, \tag{18}$$

which has the next closed form solution:

$$W_l = (f_{l-1}^A (f_{l-1}^A)^\top + f_{l-1}^B (f_{l-1}^B)^\top)^{-1} (f_{l-1}^A (f_{l-1}^A)^\top W_l^A + f_{l-1}^B (f_{l-1}^B)^\top W_l^B). \tag{19}$$

Next, we wish to prove Proposition H.4.

*Proof.* Given that Git Re-Basin converged to a permutation matrix $P_l$, full-rank FS-Merge can produce the same solution using

$$M_l = \begin{pmatrix} \frac{1}{2} I & \frac{1}{2} P_l \end{pmatrix}, U_{l-1} = \begin{pmatrix} I \\ P_{l-1}^\top \end{pmatrix}. \tag{20}$$

By additionally assuming that the features $f_l^B$ have a low rank of $r$, there exist projection matrices $C_l, C_l' \in \mathbb{R}^{d \times d}$, both of rank $r$, such that:

$$P_l f_l^B = C_l' P_l C_l f_l^B \,. \tag{21}$$

And similarly for $f_{l-1}^B$. Thus, a low-rank FS-Merge can reconstruct the next solution:

$$M_l = \begin{pmatrix} \frac{1}{2}I & \frac{1}{2}C_l' P_l C_l \end{pmatrix}, \, U_{l-1} = \begin{pmatrix} I \\ (C_{l-1}' P_{l-1} C_{l-1})^\top \end{pmatrix} \,. \tag{22}$$

Which is equivalent to the Git Re-Basin solution. This is a valid solution, as the identity matrices can be reconstructed using the diagonal components $U_{l-1}^{\text{diag}}, M_l^{\text{diag}}$, while $C_l' P_l C_l, (C_{l-1}' P_{l-1} C_{l-1})^\top$ are matrices with rank $r$ or lower.

In addition, full-rank FS-Merge can produce the same solution as RegMean using

$$M_l = (f_{l-1}^A (f_{l-1}^A)^\top + f_{l-1}^B (f_{l-1}^B)^\top)^{-1} \begin{pmatrix} f_{l-1}^A (f_{l-1}^A)^\top & f_{l-1}^B (f_{l-1}^B)^\top \end{pmatrix}, \, U_{l-1} = \begin{pmatrix} I \\ I \end{pmatrix} \,. \tag{23}$$

By additionally assuming that the features $f_{l-1}^A, f_{l-1}^B$ have a low rank of $r$, a low-rank FS-Merge with rank $2r$ can also produce the solution in Eq. 23. This is because $U_{l-1}$ can be reconstructed using its diagonal component $U_{l-1}^{\text{diag}}$, and $M_l$ is a matrix with rank $2r$ or lower.

$\square$

### H.3 Proof of Lemma H.5: Expressive power case study

In the next section, we wish to prove Lemma H.5. We examine a simple scenario where traditional merging methods fail due to their reliance on basic merging rules. In contrast, we prove here that both full-rank and low-rank FS-Merge can handle this scenario.

In more detail, our scenario (Eq. 24) involves two linear models, $A$ and $B$, which generate features for two different tasks. We assume that both models share the same backbone weight matrix, $W^* \in \mathbb{R}^{d \times d}$, and that $X \in \mathbb{R}^{d \times N}$ represents the inputs. Each model then selects a subset of shared features using $V^{A/B} \in \mathbb{R}^{d \times d}$, which are diagonal matrices with 0/1 entries, each having a rank of $\frac{r}{2}$. The two matrices do not have 1s in the same positions. Next, the two models apply a permutation matrix $P^{A/B} \in \mathbb{R}^{d \times d}$ and a diagonal matrix with positive entries that scale the features, $D^{A/B} \in \mathbb{R}^{d \times d}$. The ReLU function $\sigma$ is used to generate the post-activation features $f^A, f^B \in \mathbb{R}^{d \times N}$, which are later used by the classification head to make predictions.

$$z^A = D^A P^A V^A W^* X \,, \; f^A = \sigma(z^A) \,, \;\; z^B = D^B P^B V^B W^* X \,, \; f^A = \sigma(z^B) \,. \tag{24}$$

We will also denote

$$z^* = (V^A + V^B) W^* X \,, \; f^* = \sigma(z^*) \,. \tag{25}$$

Next, we prove Lemma H.5 in Appendix H.3.1, Appendix H.3.2 and Appendix H.3.3, demonstrating that low-rank FS-Merge can achieve a zero-loss solution, while Git Re-Basin and RegMean cannot.

### H.3.1 FS-Merge

Low-rank FS-Merge is capable of reconstructing $f^*$, using

$$M = \begin{pmatrix} (P^A)^\top (D^A)^{-1} & (P^B)^\top (D^B)^{-1} \end{pmatrix}, \, U = \begin{pmatrix} I \\ I \end{pmatrix} \,. \tag{26}$$

As the obtained features will be:

$$\tilde{f} = \sigma(\tilde{z}) = \sigma\left(M\begin{pmatrix} W^A & 0 \\ 0 & W^B \end{pmatrix} UX\right) =$$

$$\sigma\left(\begin{pmatrix}(P^A)^\top (D^A)^{-1} & (P^B)^\top (D^B)^{-1}\end{pmatrix}\begin{pmatrix} D^A P^A V^A W^* & 0 \\ 0 & D^B P^B V^B W^* \end{pmatrix}\begin{pmatrix} I \\ I \end{pmatrix} X\right) = \sigma(z^*) = f^*. \quad (27)$$

Note that Eq. 26 is a valid solution, as $M$ has a rank of at most $r$, and $U$ is a concatenation of diagonal matrices, and therefore, they can be parameterized by low-rank FS-Merge with rank of $r$.

Moreover, the low-rank Foldable SuperNet can perfectly reconstruct the features of both linear layers, achieving zero loss, using

$$M = \begin{pmatrix} V^A (P^A)^\top (D^A)^{-1} & V^B (P^B)^\top (D^B)^{-1} \end{pmatrix}, U = \begin{pmatrix} D^A P^A V^A \\ D^B P^B V^B \end{pmatrix}. \quad (28)$$

Note that the operation of $D^{A/B} P^{A/B} V^{A/B}$ and $\sigma$ is commutative, and thus

$$f^A = D^A P^A V^A f^*. \quad (29)$$

The same holds for $f^B$. Using it, the Foldable SuperNet output is

$$\tilde{f}(z^A, z^B) = U\sigma\left(M\begin{pmatrix} z^A \\ z^B \end{pmatrix}\right) = \begin{pmatrix} D^A P^A V^A \\ D^B P^B V^B \end{pmatrix}\sigma\left(\begin{pmatrix}(V^A (P^A)^\top (D^A)^{-1} & V^B (P^B)^\top (D^B)^{-1}\end{pmatrix}\begin{pmatrix} D^A P^A V^A W^* X \\ D^B P^B V^B W^* X \end{pmatrix}\right)$$

$$= \begin{pmatrix} D^A P^A V^A \\ D^B P^B V^B \end{pmatrix}\sigma\left(\begin{pmatrix} V^A & V^B \end{pmatrix}\begin{pmatrix} V^A W^* X \\ V^B W^* X \end{pmatrix}\right) = \begin{pmatrix} D^A P^A V^A \\ D^B P^B V^B \end{pmatrix}\sigma(z^*) = \begin{pmatrix} D^A P^A V^A f^* \\ D^B P^B V^B f^* \end{pmatrix} = f^A \,||\, f^B. \quad (30)$$

Moreover, this is a valid solution, as both $M$ and $U$ from Eq. 28 are matrices with a rank of at most $r$, and therefore, they can be parameterized by low-rank FS-Merge with rank of $r$.

We note that full-rank FS-Merge can generate any solution obtained by low-rank FS-Merge since it can use matrices of any rank. Therefore, full-rank FS-Merge is capable of handling these scenarios as well.

### H.3.2 RegMean

Under our assumptions, RegMean's Jin et al. (2023) objective is:

$$W = \underset{W}{\operatorname{argmin}} \left\| WX - D^A P^A V^A W^* X \right\|_2^2 + \left\| WX - D^B P^B V^B W^* X \right\|_2^2. \quad (31)$$

Which uses the Frobenius norm. For RegMean to achieve zero loss, it is required that $WX = D^A P^A V^A W^* X$. Thus

$$W = D^A P^A V^A W^* X X^\dagger. \quad (32)$$

Where we denote $\dagger$ as the Penrose-Moore pseudo-inverse. Similarly, to achieve zero error, we must also ensure that $W = D^B P^B V^B W^* X X^\dagger$. Using both results we obtain:

$$D^A P^A V^A W^* X X^\dagger = D^B P^B V^B W^* X X^\dagger. \quad (33)$$

Note that Eq. 33 is a set of rational equations, and therefore it can be transformed to a set of polynomial equations. These polynomials are not identically zero, due to the assumption $r > 0$. Therefore, the set of solutions of these equations is a zero-measure set (under the Lebesgue measure). This means the chances of randomly selecting parameters (from some continuous distribution) that satisfy this equation are zero. Therefore, in general, RegMean cannot find parameters that satisfy this condition, so it lacks the expressive power to handle the case being studied.

### H.3.3 Git Re-Basin

Under our assumptions, Git Re-Basin's Ainsworth et al. (2023) objective is:

$$P = \underset{P \in S_d}{\operatorname{argmin}} \left\| D^A P^A V^A f^* - P D^B P^B V^B f^* \right\|_2^2 . \tag{34}$$

For RegMean to achieve zero loss, it is required that

$$D^A P^A V^A f^* - P D^B P^B V^B f^* = 0 . \tag{35}$$

Eq. 35 is a polynomial, and thus we can apply the same reasoning as in the previous section, leading to the conclusion that the probability of this equation holding in general is zero. Therefore, Git Re-Basin lacks the expressiveness needed to handle this case.

$\square$

