# OpenReview forum: "Foldable SuperNets: Scalable Merging of Transformers with Different Initializations and Tasks"
_TMLR — Accepted by TMLR_

### Review · Reviewer_JgDq · 2025-05-19

**Summary Of Contributions:**

The paper proposes a novel and efficient technique for merging neural networks with various initializations and tasks, referred to as Foldable SuperNets (FS-Merge). FS-Merge aims to merge individually trained massive transformer models as its primary objective, exceeding the limitation of existing methods based on shared initialization. The paper addresses the challenging scenario where models trained under various tasks from diverse initializations must be merged effectively without any performance loss. The new FS-Merge method does this by creating a Foldable SuperNet that wraps the original models with frozen weights and trains a feature reconstruction goal to best combine them. The combined model is then folded back down to the original model size, maintaining computational efficiency comparable to Knowledge Distillation (KD) while improving expressiveness and data efficiency by a significant margin.

The key contributions are:

1. Introduction of the FS-Merge framework to merge transformers with different initializations and tasks efficiently.

2. Creation of the Foldable SuperNet method that inserts models with frozen weights and trains using a reconstruction loss.

3. Empirical validation on a variety of architectures, sizes, tasks, and modalities, with SOTA performance, especially in low-data regimes.

4. Theoretical validation of FS-Merge being more expressive than conventional merging techniques.

5. Demonstration of robustness in diverse settings, including vision and text transformers, highlighting the method's adaptability.

**Audience:**

Yes

**Claims And Evidence:**

Yes

**Requested Changes:**

1. Expand the scalability analysis to include combining larger quantities of models, especially in practical settings where multiple models are combined.

2. Add an ablation study on initialization methods to see how they impact combining quality more profoundly.

3. Offer more understandable descriptions during the theoretical sections to enhance content readability by readers with diverse mathematical backgrounds.

4. Compare the computational cost of FS-Merge with other merging methods when expanding the number of models.

**Strengths And Weaknesses:**

# Strengths:

1. FS-Merge addresses a significant gap in model merging by making it possible to merge models from different initializations, which is a practical and challenging problem.

2. The method is computationally efficient and data-efficient and therefore can be extended to real-world applications where large training data availability is scarce.

3. The paper provides strong empirical evidence by performing experiments on a wide range of neural architectures and tasks, showing the efficacy of the proposed method.

4. The theoretical analysis is sound and forms a good foundation for the technique, showing that FS-Merge is more expressive than traditional methods.

5. The flexibility of the suggested method in handling different model architectures (e.g., MLPs, ViTs, and BERTs) is a strong point.

6. The thorough comparison with baseline methods, including averaging, alignment, and distillation-based methods, is comprehensive and easy to understand.

# Weaknesses

1. The scalability of the method to a large number of models is discussed but not fully evaluated in large-scale settings.

2. Although FS-Merge outperforms KD and standard merging methods, computational costs in merging large sets of models can be further investigated.

3. The performance may be affected by the initialization method of FS-Merge, and a more thorough ablation on initialization methods would be beneficial.

---

> ### Author Response · Authors · 2025-06-06
> **Authors' Official Comment**
>
> We sincerely appreciate the reviewer’s insightful comments and suggestions, which have helped us refine our work. We have added  new experiments requested by the reviewer to our updated manuscript.
>
> > 1. “Expand the scalability analysis to include combining larger quantities of models, especially in practical settings where multiple models are combined”.
>
> Thank you for this important suggestion. As requested, we added new experiments to our manuscript (Appendix F.6), where we merged all our existing ViT-B-16 models, each trained from a different initialization on a different task, a total of eight models. We used a relatively small amount of data: 100 original images and 1000 augmented images per task. The models were trained on Cars, DTD, CIFAR100, GTSRB, RESISC45, MNIST, EuroSAT, and SVHN, which together comprise around 5000 classes. Under this extremely challenging setup, our method outperformed the baselines and even achieved higher joint accuracy than the ensemble.
>
> > 2. “Add an ablation study on initialization methods to see how they impact combining quality more profoundly.”
>
> We note that our manuscript includes an ablation study examining the effect of different initializations on both FS-Merge (Appendix G.1) and KD (Appendix G.2). Our experiments show that, for both methods, initializing the merged model from one of the models to be merged ("first" initialization) results in the best accuracy. We note that these appendices are already referenced in the main text (page 5).
>
> > 3. “Offer more understandable descriptions during the theoretical sections to enhance content readability by readers with diverse mathematical backgrounds”.
>
> Thank you for this helpful advice. We have revised the manuscript accordingly; the changes are highlighted in blue.
>
> > 4. “Compare the computational cost of FS-Merge with other merging methods when expanding the number of models”.
>
> A comparison and discussion on this matter have been added to our updated manuscript (Appendix F.6).

---

### Review · Reviewer_pvuc · 2025-05-26

**Summary Of Contributions:**

This work addresses the challenging task of merging transformer models trained on different tasks with distinct initializations, where traditional merging methods often fail. The authors propose **Foldable SuperNet Merge (FS-Merge)**, a novel technique that embeds the original models with frozen weights into a SuperNet, which is trained using a feature reconstruction loss and then folded into a single model of standard size. FS-Merge is extended to support the complexity of transformers and is further scaled through FS-Merge Seq., a sequential variant that reduces computational overhead. The method is theoretically shown to be more expressive than prior approaches and achieves superior empirical performance across MLPs, Vision Transformers, and Text Transformers, particularly in low-data and multi-task scenarios.

**Audience:**

Yes

**Claims And Evidence:**

No

**Requested Changes:**

1. **Add missing state-of-the-art baselines (Critical)**: Compare FS-Merge to 2024 weight-merging / PEFT methods such as TIES-Merging, AdaMerging, Task-Arithmetic 2.0, BitFit/Adapter fusion and Rank-Free SLERP so the SOTA claim is substantiated.

2. **Clarify or relax the “no-data” assumption (Critical)**:  State explicitly that FS-Merge needs *unlabeled* task data **or** demonstrate a variant that works with zero data / public proxy data to align with the stated motivation.

3. **Extend the theoretical guarantee to transformers (Critical)**: Either prove expressivity/convergence for attention blocks or clearly bound the limitation to MLPs.

4. **Equalise hyper-parameter budgets across methods (Critical)**: Cap search trials identically (or use a common auto-tuner) and document settings in the main text.

5.  **Scale experiments to 10 + merged models (Would strengthen)**: Merge a larger set of vision or language tasks and plot accuracy versus number of models.

6. **Broaden robustness testing (Would strengthen)**: Add ImageNet-C, Stylized-ImageNet and language perturbation suites; compare to all baselines.

7.  **Use trainable or merged classification heads (Would strengthen)**: Replace frozen CLIP label embeddings in at least one experiment; discuss the effect.

8. **Address depth-mismatched models (Would strengthen)**: Provide possible solution or emphasise this limitation in the conclusion.

**Strengths And Weaknesses:**

# Strengths
- FS-Merge is well-motivated, with clear architectural design choices for both MLPs and transformers. The use of a feature reconstruction loss and learnable compression/reconstruction modules is elegant and justified.
- The paper includes a meaningful theoretical result showing that FS-Merge generalizes and subsumes existing methods (e.g., RegMean, Git Re-Basin) under appropriate assumptions. This enhances the credibility of the method's expressivity claims.
- The experiments are extensive and cover multiple settings: MLPs on MNIST/Fashion-MNIST, ViTs on nine image classification datasets, and BERT on GLUE tasks.
- FS-Merge Seq. is a thoughtful engineering adaptation that addresses the growing resource cost when merging many large models. This improves the method’s applicability in real-world systems.
- Unlike prior work that assumes a common pre-trained initialization, FS-Merge handles models trained from entirely distinct random seeds and fine-tuned on different tasks.
- To keep the number of learnable parameters linear in layer width (rather than quadratic), the paper adopts a LoRA-style decomposition (low-rank plus diagonal) for the merging matrices.
# Weaknesses

1. **High computational and memory requirements:** Despite being more expressive, FS-Merge (and even KD) is significantly more expensive than simple averaging or alignment methods. Although the sequential variant (FS-Merge seq.) alleviates this somewhat, it still incurs substantial resource use and introduces a small accuracy drop (Table 7).

2. **Dependence on availability of unlabeled data**: FS-Merge requires access to “a small unlabeled subset of the original training data,” which may not be available in privacy-sensitive or proprietary settings—undermining its applicability when no data can be shared

3. **Architecture constraints (identical depths/widths only)**: The method only handles models of the same architecture and width. “One cannot naively merge two models of different depths using our method,” and no solution is provided. Thre  authors list this as future work, yet many practical cases (e.g., ViT-B with ViT-S) are depth-mismatched.

4. **Lack of theoretical guarantees for transformers**: The expressive-power result (Theorem 2.1) is stated only for MLPs (“FS-Merge can exactly reproduce any target MLP model…”), with no formal counterpart for transformers—leaving its theoretical foundations in the transformer setting unaddressed

5. **Limited evaluation scope**: Experiments focus on small-scale classification splits (e.g., MNIST halves, 9 vision tasks with ViT-B/16, GLUE pairs). There is no evaluation on larger, more realistic multi-task benchmarks (e.g., many simultaneous tasks or production-scale multitask settings). Further, the paper compares only to simple averaging, SLERP, RegMean, “Opt,” and KD. It omits several recent, advanced alignment or merging techniques (e.g., REPAIR, equivariant weight alignment) that could be competitive (Related Work, Appendix A.1)

6. **Hyper-parameter tuning is heavy** and carried out on validation data, risking unfair advantage over baselines. Grid search over 40+ settings per method (detailed in Appendix E.2) versus single default settings for others like SLERP/average.

7. **Hyperparameter sensitivity and tuning complexity**:  FS-Merge introduces many hyperparameters (learning rates, ranks r, initialization schemes), but the main text offers almost no guidance on their selection or sensitivity; all details are relegated to the appendix (Appendix E).

8. **Sequential merging trade-offs underexplored**: While FS-Merge seq. reduces memory, it is only briefly described and empirically shown to incur a minor accuracy drop. The paper does not analyze how the order of sequential merges affects final performance or whether better schedules exist (§2.2 and Appendix B.4).

9. **Task Head Assumption**: FS-Merge assumes task-specific classification heads remain separate, which simplifies merging but and may inflate reported gains. Further, it limits end-to-end deployment for some multitask systems (real multi-task setups normally train or merge the heads). A discussion on merging or adapting classification heads would be valuable.

10. **High Cost of Full-Rank M and U** While theoretically more expressive, using full-rank M and U matrices in the ViT case is not recommended due to very high memory and time complexity.

11. **Simplicity of the Method for Transformers**: While the authors state "FS-Merge is simple", and the core concept is indeed straightforward, its application to complex architectures like transformers (Figure 3, Appendix B.1 ) is quite involved with multiple M and U matrices at different stages of the attention and MLP blocks.

In summary, most prominent messages in this paper: simplicity, SOTA status, theoretical guarantee for transformers, and robustness/generalisation advantages—are not backed by sufficiently comprehensive or clear evidence. Additional experiments, broader baselines, quantitative cost analysis and an extended proof are required for the evidence package to be convincing.

---

> ### Author Response · Authors · 2025-06-06
> **Authors' Official Comment - Part 1**
>
> We sincerely appreciate the reviewer’s thoughtful feedback. We have revised our manuscript and added additional experiments as requested.
>
> > 1. “Compare FS-Merge to… TIES-Merging, AdaMerging, Task-Arithmetic 2.0, BitFit/Adapter fusion and Rank-Free SLERP”.
>
> We searched but could not find any methods with the names “Task-Arithmetic 2.0” and “Rank-Free SLERP”. Could the reviewer kindly refer us to these works?
>
> Regarding BitFit [1] and AdapterFusion [2], please note that we already compared against LoRA, a stronger, more recent, and widely adopted PEFT method. As demonstrated, FS-Merge consistently outperforms this SOTA baseline (Tables 3, 4).
>
> As for TIES-Merging [4] and AdaMerging [5], these approaches are incompatible with our setting, as they explicitly assume that the models were fine-tuned from a common pre-trained model. Moreover, these traditional methods, like the Average and RegMean baselines, are efficient but have extremely limited expressivity, making them too weak for our setting.
>
> To further address this concern, we applied TIES-Merging to the experiments in Table 3. Like the other traditional merging methods, it resulted in a merged model that performed no better than random guessing.
>
> TIES-Merging:
>
> DTD, EuroSAT: Per-task accuracy 7.62 | Joint accuracy 1.13
>
> CIFAR100, SVHN: Per-task accuracy 5.88 | Joint accuracy 1.02
>
> RESISC45, SVHN: Per-task accuracy 6.04 | Joint accuracy 2.76
>
> > 2. “Clarify or relax the “no-data” assumption… State explicitly that FS-Merge needs unlabeled task data”.
>
> We would like to highlight that we never claimed that our method requires no data. On the contrary, we emphasize multiple times throughout the manuscript that FS-Merge requires unlabeled task data. Below are just a few examples from the main text that illustrate this point:
>
> Page 2: “Like other methods (Jin et al., 2023; Ainsworth et al., 2023), it requires only an unlabeled fraction of the original training data”.
>
> Page 3: “using unlabeled subsets of each task’s training set, D^A and D^B...”.
>
> Page 7: “When merging the models, a small unlabeled subset of their original training data is used”.
>
> Page 11: “Limitations. FS-Merge requires a small unlabeled subset of the original training data, similarly to most previous merging methods”.
>
> > 3. “Either prove expressivity/convergence for attention blocks or clearly bound the limitation to MLPs”.
>
> Thank you for this suggestion. To address this concern, we highlighted this limitation in the abstract and on page 6 of our updated manuscript.
>
> This expressiveness result holds for most Transformer components, as many of them are MLP-based. We believe the main challenge is to extend this proof to the attention mechanism. Currently, we use a shared U matrix for keys, values, and queries, rather than a separate one for each. We leave this extension for future work.
>
> > 4. “Equalise hyper-parameter budgets across methods”.
>
> As shown in Appendix E.1 and Appendix E.2, we used the exact same hyperparameter grid for FS-Merge, Knowledge Distillation, and the LoRA baselines. Regarding baselines such as RegMean and Opt, these methods have relatively few hyperparameters to begin with, and we performed an exhaustive search over the available ones. The failure of these methods in our setting is not due to the choice of hyperparameters, but rather because they lack the expressiveness needed to handle this challenging scenario.
>
> > 5. “Scale experiments to 10 + merged models”.
>
> Thank you for this important proposal. As requested, we added new experiments to our manuscript (Appendix F.6), where we merged all our existing ViT-B-16 models, each trained from a different initialization on a different task, a total of eight models. Under this extremely challenging setup, our method outperformed the baselines and even achieved higher joint accuracy than the ensemble.
>
> > 6. “Broaden robustness testing”.
>
> Thank you for this suggestion. We do aim to broaden our robustness evaluation in future work.
>
> > 7. “Use trainable or merged classification heads”.
>
> In Appendix F.7, we merge BERT models trained from different initializations and tasks, without freezing the classification heads. We have clarified this setting in the updated manuscript.
>
> > 8. “Address depth-mismatched models (Would strengthen): Provide possible solution or emphasise this limitation in the conclusion.”
>
> We would like to point out that this limitation is already discussed in the summary section, under the "Limitations" paragraph (page 11). We consider this an important and interesting direction, and we plan to investigate it further in future work.
>
> [1] BitFit: Simple Parameter-efficient Fine-tuning for Transformer-based Masked Language-models
>
> [2] AdapterFusion: Non-Destructive Task Composition for Transfer Learning
>
> [3] LoRA: Low-Rank Adaptation of Large Language Models
>
> [4] TIES-Merging: Resolving Interference When Merging Models
>
> [5] AdaMerging: Adaptive Model Merging for Multi-Task Learning

---

> ### Author Response · Authors · 2025-06-06
> **Authors' Official Comment - Part 2**
>
> We would also like to address several points raised in the weaknesses section.
>
> > 1. “Despite being more expressive, FS-Merge (and even KD) is significantly more expensive than simple averaging or alignment methods”.
>
> As noted in the article (e.g., Table 1), traditional efficient merging methods fail catastrophically in our challenging setting of merging transformers with different initializations. Consequently, only resource-intensive methods are effective in this scenario.
>
> > 3. “The method only handles models of the same architecture and width”.
>
> Please note that, in contrast to methods such as Average, RegMean, and task vectors, our method is capable of merging models with different widths. We also mention this in our manuscript: “FS-Merge can be easily extended to merge any number of models or models with varying widths”, (page 3). This can be achieved by simply concatenating features of different widths and adjusting the sizes of M and U accordingly. We plan to explore this direction further in future work.
>
> Regarding the same-architecture assumption: indeed, most merging methods, including ours, rely on this assumption. While this does limit the applicability of many merging approaches, it is important to note that the Transformer architecture has become extremely popular and is widely used across a broad range of tasks. Thus, merging models with the same architecture remains an extremely important and practical goal, as demonstrated by the growing interest in the model merging field.
>
> > 5. ”It omits several recent, advanced alignment or merging techniques (e.g., REPAIR, equivariant weight alignment) that could be competitive”.
>
> We wish to emphasize that both REPAIR [6] and Equivariant Weight Alignment [7] are alignment-based methods originally introduced for CNNs, not for transformers, and were therefore not included in our comparisons. Moreover, Opt can be seen as a generalization of alignment methods, as it enables “soft” alignment. Since Opt was not expressive enough to handle our challenging setting, it is unlikely that REPAIR or Equivariant Weight Alignment would perform better.
>
> [6] REPAIR: REnormalizing Permuted Activations for Interpolation Repair.
>
> [7] Equivariant Deep Weight Space Alignment.

---

### Review · Reviewer_kq3s · 2025-05-27

**Summary Of Contributions:**

This is an interesting work. It targets a harder problem of merging large Transformer-based models trained on different tasks from distinct initializations. Traditional merging suffers from catastrophic forgetting. Knowledge distillation achieves much better results with the price of a higher cost. In light of this view, the authors propose FS-Merge.

**Audience:**

Yes

**Claims And Evidence:**

Yes

**Requested Changes:**

Please see above weaknesses.

**Strengths And Weaknesses:**

Strength:
The motivation of this paper is sound, the paper experimentally shows that current approaches are insufficient.
The experiments are mostly comprehensive, see weakness for additional comments.

Weakness:
1. The implementation of KD needs further clarification. For example, is the KD here supervises only the final layer. Currently, there are lots of approaches that exceed the final layer supervision (e.g., intermediate layer/feature supervision [1-3]. They should be in detailed discussed and highlight the novelty of the proposed method). Some of them are specifically designed for Transformer-based architectures.
2. I notice a recent paper mentioning language-based Transformer fusion [4], please highlight the technical differences between them.

[1] Universal-kd: Attention-based output-grounded intermediate layer knowledge distillation

[2] Ad-kd: Attribution-driven knowledge distillation for language model compression

[3] AMD: Automatic Multi-step Distillation of Large-scale Vision Models

[4] Knowledge Fusion of Large Language Models

---

> ### Author Response · Authors · 2025-06-06
> **Authors' Official Comment**
>
> We deeply appreciate the time and effort the reviewers have dedicated to reviewing our work and thank them for the constructive feedback.
>
> > 1. “The implementation of KD needs further clarification. For example, is the KD here supervises only the final layer. Currently, there are lots of approaches that exceed the final layer supervision (e.g., intermediate layer/feature supervision [1-3]. They should be in detailed discussed and highlight the novelty of the proposed method)”.
>
> We appreciate this important suggestion. Indeed, we have evaluated several Knowledge Distillation variants that aim to reconstruct the internal features of the transformers, experimenting with different regression loss functions and various choices of layers from which features are extracted for supervision. We report several of these experiments in Appendix G.5. Additionally, we tested KD variants that initialize the student model using different merging methods and then continue training it (Appendix G.2). We found that our strongest KD variant, where the student is initialized from one of the models to be merged and trained to mimic the last-layer features of all models, outperforms all these baselines. We clarified the implementation details of our KD baseline in the final manuscript (page 8), added a reference to the experiments on different KD variants, and cited [1–3] (page 44).
>
>
> > 2. “I notice a recent paper mentioning language-based Transformer fusion [4], please highlight the technical differences between them”.
>
> Thank you for this suggestion. Please note that the setting in [4] is different from ours. [4] aims to fuse the knowledge of multiple generative LLMs with varying architectures **on the same task**. In contrast, our method focuses on classification models that share the same architecture but **are trained on different tasks**.
>
> Specifically, [4] generates logits from multiple LLMs on the same text corpus, merges these logits, and uses them to transfer knowledge into a single LLM (“Consequently, different probabilistic distribution matrices for the same text, originating from various LLMs, can be used to represent the diverse knowledge embedded within these models” [4]). This method assumes that all the original models have the ability to solve this single task.
>
> Therefore, [4] would not enable knowledge merging in our setting, where each model was trained on a specific task that the other models are not capable of solving. Creating logits from all these models on the same task would not allow us to transfer their knowledge into the merged model.
>
> We have added this reference and the corresponding discussion to the updated manuscript (page 20).
>
> [1] Universal-kd: Attention-based output-grounded intermediate layer knowledge distillation
> [2] Ad-kd: Attribution-driven knowledge distillation for language model compression
> [3] AMD: Automatic Multi-step Distillation of Large-scale Vision Models
> [4] Knowledge Fusion of Large Language Models

---

### Decision · Action_Editor_823g · 2025-07-07

**Recommendation:** Accept as is

**Audience:**

Yes

**Audience Explanation:**

The area of model merging is increasing in significance with the plurality of pre-trained models available. This paper could therefore potentially interest a sizeable part of the community.

**Claims And Evidence:**

Yes

**Claims Explanation:**

This paper explores model merging in a relatively unexplored setting: combining large transformers trained on different tasks from distinct initializations.

Reviewer **kq3s** notes that “the motivation is sound,” the experiments are “mostly comprehensive,” and that the paper “experimentally shows that current approaches are insufficient.” They highlight some missing knowledge distillation (KD) references, which the authors added, and acknowledge that the KD baselines used are strong.

Reviewer **pvuc** finds the method “well-motivated,” with “clear architectural design choices.” They note that the paper presents a “meaningful theoretical result showing that FS-Merge generalizes and subsumes existing methods under appropriate assumptions.” While they listed several weaknesses, these were addressed by the authors in the revision.

Reviewer **JgDq** agrees that the method “addresses a significant gap in model merging” and “provides strong empirical evidence” by evaluating across a wide range of architectures and tasks. They consider the theoretical analysis sound and the comparisons thorough and comprehensive. The weaknesses they pointed out were addressed with additional experiments and discussion.

All three reviewers are **leaning towards acceptance** following the revisions.

The AC agrees that the paper tackles an important and underexplored topic and offers valuable contributions to the community. There is no reason to overturn the unanimous recommendations in favor of acceptance.